# Spatial distribution of groundwater recharge, based on regionalized soil moisture models in Wadi Natuf karst aquifers, Palestine

**Clemens Messerschmid[1], Amjad Aliewi[2]**

[1] Hydrologist, Independent researcher, Ramallah, Palestine

[2] Water Research Center, Kuwait, Kuwait Institute for Scientific Research, Kuwait

**Correspondence:** Clemens Messerschmid (clemensmesserschmid@yahoo.de)

**Abstract.**

While groundwater recharge is considered fundamental to hydrogeological insights and basin management and studies on its temporal variability amass, much less attention has been paid to its spatial distribution, by comparison. And in ungauged catchments it has rarely been quantified, especially on the catchment scale.

For the first time, this study attempts such analysis, in a previously ungauged basin. Our work based on field data of several soil moisture stations, which represent five geological formations of karst rock in Wadi Natuf, a semi-arid to sub-humid Mediterranean catchment in the occupied Palestinian West Bank. For that purpose, recharge was conceptualized as deep percolation from soil moisture under saturation excess conditions, which had been modelled parsimoniously and separately with different formation-specific recharge rates.

For the regionalisation, inductive methods of empirical field-measurements and observations were combined with deductive approaches of extrapolation, based on a new basin classification framework (BCF) for Wadi Natuf, thus following the recommendations for hydrological Prediction in Ungauged Basins (PUB), by the International Association of Hydrological Sciences (IAHS). Our results show an average annual recharge estimation in Wadi Natuf Catchment (103 km$^2$), ranging from 235 to 274 mm (24 to 28 Mm$^3$) per year, equivalent to recharge coefficients (RC) of 39-46% of average annual precipitation (over a 7-year observation period but representative for long-term conditions as well).

Formation-specific RC-values, derived from empirical parsimonious soil moisture models, were regionalised and their spatial distribution was assessed and quantified on the catchment scale. Thus, for the first time, a fully distributed recharge model in a hitherto entirely ungauged (and karstic) aquifer basin was created that drew on empirical methods and direct approaches. This was done by a novel combination of existing methods and by creating a unified conceptual basin classification framework for different sets of physical basin features. This new regionalisation method is also applicable in many comparable sedimentary basins in the Mediterranean and worldwide.

**Keywords.** Distributed recharge, classification framework, regionalisation PUB, landscape features

## 1 Introduction

The assessment of distributed groundwater recharge is considered a challenge already in basins with scarce data; even more so its spatial distribution and the regionalisation of point measurements and plot-scale experiments, since the governing processes of recharge and its spatial distribution often remain poorly understood (Hartmann *et al.*, 2012a) even in well-developed basins. An additional complication poses the nature of karstic aquifers, characterized by their diverse and complicated inhomogeneous and anisotropic flow fields (Schmidt *et al.*, 2014 and Geyer *et al.*, 2008). Yet, regionalised information on spatially distributed recharge is highly important, not only for the correct budgeting of inflows on different scales, but also for the overriding and growing demands in resource protection and sustainable management, as well as the equitable allocation of groundwater among different basin riparians.

## 1.1 Spatial variability in ungauged basins and physical landscape characteristics

In order to investigate spatial variability, two shifts in general approaches can be observed; on the one hand a shift from so-called indirect to direct approaches, which try to observe, determine and quantify surface-near processes (Dörhöfer and Jesopait, 1997; Lerner *et al.*, 1990), as discussed in Messerschmid *et al.* (2020). This is particularly the case in areas, where the observation of deep underground surfaces is limited or severely restricted, as in Wadi Natuf, where the Israeli occupation prohibits any well development (World Bank, 2009). And on the other hand, many authors of the PUB-literature recommended a shift away from lumped and integrated models (such as Richts *et al.*, 2011 or MacDonald *et al.*, 2021), which may be problematic in ungauged basins, especially where lateral flow connections to other basins exist. Instead, distributed models are recommended that differentiate hydrological processes, such as soil saturation, runoff or recharge, together with their drivers, e.g. precipitation and evapotranspiration (Batelaan and de Smedt, 2001, 2007; Hrachowitz *et al.*, 2013 and Sivakumar *et al.*, 2013).

To constrain parameters and to ensure that hydrological models faithfully and realistically represent the dominant processes, highly location-specific, empirical work should be combined with conceptual efforts, such as the correct differentiation of different groups of landscape features that rule the recharge process (Franchini and Pacciani, 1991).

The processes that enable and limit recharge, besides precipitation, encompass evapotranspiration, runoff, soil infiltration and deep percolation. Direct evaporation from the surface largely depends on the balance of soil moisture storage (see Messerschmid *et al.*, 2020), whereas transpiration reflects the type of vegetation (type and density of plant cover). Surface runoff is a function of landscape characteristics (slope, vegetation, land use, etc.) and in addition, infiltration excess runoff is directly linked to the am SM storage as well as the permeability of the underlying rock. Soil infiltration can depend on soil types and other factors, such as vegetation. Finally, percolation into the rock formation is closely linked to the mineralogical content of the rock, its permeability and conductivity (here combined as *geology*). In other words, most processes are ruled by three sets of physical catchment characteristics: Geology (1), soil (2) and land forms or features of land use and land cover (LU/LC) (3).

According to Hrachowitz *et al.* (2013), most studies select parameter sets from these three principal groups of physical characteristics or their combination and interaction. Sanz *et al.* (2011) used geology and lithology (first group). Batelaan and de Smedt (2001), Batelaan and de Smedt (2007) and Aish, Batelaan and de Smedt (2010) used soil characteristics (second group) and combined them with land use, topography, water level data and lithology. Aish, Batelaan and de Smedt (2010) and Zomlot *et al.* (2015) used landscape features, including topography, vegetation and land use (third group), which can be combined to the above land use and land cover characteristics (LU/LC). Finally, some authors use a selection of many parameters, based, however not on field observations but on conceptual assumptions (Radulovič *et al.*, 2011) in order to assign them with weights as variables in a basin-wide transfer function between spatial characteristics and hydrological response or in order to estimate the relative importance of different "conditioning factors" (Jaafarzadeh *et al.*, 2021).

The linkage between physical characteristics and hydrological processes, between catchment form and function (Hrachowitz *et al.*, 2013) can be done by so-called catchment classification and similarity frameworks, based on field observations and on similarities of hydrological function (McDonnell and Woods, 2004; Berne *et al.,* 2005). This is best done on the catchment scale, at which the entire complexity of distributed recharge processes and their interactions is fully at play (Hrachowitz *et al.*, 2013; McDonnell *et al.*, 2007). By contrast, studies on a very small scale - continent wide (MacDonald *et al.*, 2021) or even global (Richts, *et al.*, 2011; Mohan *et al.*, 2018) cannot live up to this demand.

According to Sivapalan *et al.* (2003a), such predictive systems should contain three components – (1) a model that describes key processes, (2) climatic input with the meteorological drivers of basin response and (3) parameters of landscape properties that govern these processes. In other words, basin classification frameworks differentiate, describe and, where possible, quantify the observable physical landscape features, both underground (using geology) and above surface (using soil cover or land use and land cover, LU/LC) and relate them to each other.

For their part, Sivakumar *et al.* (2013) offer a three-step procedure for an effective formulation and verification of a catchment classification framework: (1) the detection of possible patterns in hydrologic data and determination of complexity and connectivity levels; (2) the classification into groups and subgroups based on data patterns, system complexity and connections; and (3) the verification of the classification framework.

For the regionalisation of physical parameters, in ungauged basins, previous authors have suggested the use of physiographic similarity as a proxy (Arheimer and Brandt, 1998; Parajka *et al.*, 2005; Dornes *et al.*, 2008; Masih *et al.*, 2010). However, the correct linkage and translation of point- and plot-scale observations into regionalised findings on the catchment scale often remains a crucial challenge (Hartmann *et al.*, 2013). And the regionalisation of observable spatial parameters remains connected to the empirical efforts of field observation and measurements (maps, aerial photography, satellite imagery and of course field visits). This article therefore draws on the recharge measurements and modelling in Messerschmid *et al.* (2020).

**1.2 Reliable field data in regional flow systems for the correlation of form and function**

Several factors pose a challenge to the estimation of spatially distributed recharge, such as the need for reliable field data or the correct conceptual representation of the aquifer and its flow and recharge processes (Goldscheider & Drew, 2007; Scanlon *et al.*, 2006). Important physical landscape features are often difficult to control since they are spatially highly variable, localised in nature and far from being uniquely correlated to each other (Beven, 2000; Oudin *et al.*, 2010). Yet, they shape the overlapping processes of groundwater recharge (Batelaan and De Smedt, 2001; Beven and Kirkby, 1979).

The translation of physical basin form into hydrological function is crucial and challenging, since it involves two discrete conceptual levels and an extraordinary complexity of interactions. Therefore, many studies suggested the use of physiographic similarity as a proxy for functional similarity (Arheimer and Brandt, 1998; Parajka *et al.*, 2005; Dornes *et al.*, 2008; Masih *et al.*, 2010). Hence, the regionalisation of runoff, recharge or other dynamic catchment response characteristics can be based on physical characteristics (Yadav *et al.*, 2007).

Hydrological system signatures, e.g. temporal patterns discharge, flow duration curves or spring hydrographs, can be employed to create a link between physical features and basin response and to describe emergent system properties (Eder *et al.*, 2003; Hartmann *et al.*, 2013). They can be used quantitatively, e.g. for the calibration of models (Hingray *et al.*, 2010; Baalousha *et al.*, 2018), or qualitatively, as indicators of basin response (see Messerschmid *et al.*, 2020; Sivapalan *et al.*, 2003b and Winsemius *et al.*, 2009). And in poorly gauged catchments, they can serve the regionalisation of plot-scale findings into basin-wide overall processes (e.g. Castellarin *et al.*, 2004; Bulygina *et al.*, 2009 and Pallard *et al.*, 2009), or the testing and investigation of modelling results (see Messerschmid *et al.*, 2020).

Field observations can either focus on surface water or on the saturated, deeply buried zone of the aquifer. Where both are not available (as in Messerschmid *et al.*; 2020), the unsaturated zone can be targeted – within the soil cover or underlying rock formations – as Scanlon *et al.* (2006) report.

The problems with the regionalisation of distributed recharge are further exacerbated in deeply buried karstic aquifers, known for their non-Darcian flows and anisotropic natural flow fields, which often are further altered and disturbed by human intervention, e.g. well abstractions. In such basins, even a well-controlled basin response based on lumped outflows in the often strongly confined downstream area often does not truly reflect upstream variability in unconfined and outcropping areas, where recharge takes place. This is especially the case in settings where several sub-units are stacked and hydraulically interconnected in one uniformly acting regional aquifer in the downstream abstraction zone, with low gradients and excessive pumping, as Dafny (2009) and Dafny *et al.* (2010) have shown for the Western Aquifer Basin (see also Hartmann *et al.*, 2012b; Guttman & Zukerman, 1995 and Abusaada, 2011). Last not least, with respect to spatially distributed recharge from different hydrogeological units, the concept of budgeting in- and outflows only functions correctly when no downward leakage has to be accounted for. This process, however, is often beyond the reach of observations and measurements, particularly in poorly or entirely ungauged basins around the world.

Sivapalan *et al.* (2003a) stated that in ungauged basins predictive systems must be inferred from direct field observation of dominant processes and empirically derived field parameters. They must be firmly based on local knowledge of the observable landscape (and climate) controls of hydrological processes (see also Messerschmid *et al.*, 2020). On the other hand, McDonnell *et al.* (2007) argued that any mapping or characterization of landscape heterogeneity and process complexity must be driven by a desire to generalize and extrapolate observations from one place to another, or across multiple scales. A certain degree of extrapolation is therefore inevitable when attributing physical features and feature ensembles to processes and basin responses (or from the observed to another location). This therefore involves deductive steps. But the need for direct observation remains. PUB theory therefore postulates the imperative of a combination, or better the integration of inductive (experimental and empirical) and deductive approaches in regionalisation (Pomeroy, 2011).

Conceptually, Sawicz *et al.* (2011) developed a simple cooking recipe for regionalisation consisting of three steps: (1) classification (to give names), (2) regionalisation (to transfer information), and (3) generalization (to develop new theory), or in brief terms: name it, attribute it, theorize it.

### 1.3 Western Aquifer Basin – overview and existing recharge studies

Details of the characteristics of the Western Aquifer Basin (WAB) were described in Messerschmid *et al.* (2020). The WAB is a complex of up to 1000 m thick Upper Cretaceous carbonatic karst aquifer sequences (SUSMAQ, 2002) and conventionally divided into two regional aquifer layers (Fig. 1) – an Upper Aquifer (UA) of Turonian to Cenomanian age and a Lower Aquifer (LA) of Upper Albian age, (see Fig. 2a in Messerschmid *et al.*, 2020) – and separated by poorly or non-permeable layers of Lower Cretaceous age (Yatta formation). However, this simplified regional hydrostratigraphy applies only to the Coastal Plain downstream, with its productive abstraction and discharge zone, where the fully confined aquifer acts uniformly and with a low hydraulic gradient (Dafny *et al.*, 2010); see Table 1, section 2. On a local scale, especially in the phreatic zone upstream, the hydrostratigraphy is far more complex than the above-mentioned bipartite division into Upper and Lower Aquifers.

Importantly, whereas the productive Coastal Plain (inside Israel) is well developed, monitored and gauged through a network of hundreds of Israeli groundwater abstraction and monitoring wells (tapping the deep aquifers). By contrast, the slopes, the WAB recharge and accumulation zones (in the occupied West Bank) remain almost untouched, ungauged and unexplored, due to severe Israeli restrictions on Palestinian water use and development (World Bank, 2009) – one of the main points of contention and water conflict between the two sides. Wadi Natuf, the study area of this paper, lies almost entirely within the aquifer's recharge zone, with only the most downstream western portion bordering on the productive abstraction zone in the coastal plain (Fig. 1) and with one single abstraction well not far from the western catchment boundary.

So far, only a few authors have attempted the analysis of fully distributed recharge in the WAB (Hughes *et al.*, 2008) and no previous study was based on empirical field evidence, measurements and observations. Sheffer (2009) introduced a semi-distributed, partially lumped recharge model, however with a very coarse lithological differentiation into merely two types of rock, either permeable or less permeable. In addition, Sheffer (2009) took his soil model parameters from the general literature and later adjusted them by calibration. In his own words, he focussed and aimed at '*the understanding of temporal influence on recharge processes*', rather than on understanding spatial influences (Sheffer *et al.*, 2010; Sheffer, 2009).

During the last two decades, other studies of field-based and empirical investigations on sub-catchment, local and plot-scales were conducted. Chloride mass balance calculations were carried out in the adjacent Eastern Aquifer Basin (EAB) (Marei *et al.*, 2010; Schmidt *et al.*, 2013; Aliewi et al, 2021) and in the central WAB (Jebreen *et al.*, 2018), usually with annual RC values between 30 and 50% (see Messerschmid *et al.*, 2020, Appendix H). However, they contributed little to the spatial differentiation of distributed recharge processes, let alone, its regionalisation.

**1.4 Research gaps, aims and motivation of our study**

In the WAB, lumped studies of basin-wide replenishment are widely available, however, mostly based on desktop work. By contrast, distributed recharge quantification has hardly been attempted. Moreover that, the physical form and the spatially variable parameters that rule the recharge process were not observed or measured directly in the field. At most, some empirical recharge studies were conducted on the point scale but without further regionalisation efforts – a crucial difference according to Martínez-Santos and Andreu (2010). Under such circumstances, the regionalisation of the observed and modelled field results must include at least some measure of extrapolation and deduction. A suitable basin classification framework (BCF) for the WAB, as recommended by Hrachowitz *et al.*, (2013) above, did not exist prior to this study and had to be developed, drawing on the three groups of physical characteristics above.

The previous paper (Messerschmid *et al.*, 2020) had been firmly grounded in field observation, measurements and a forward-calculating location-specific soil-moisture percolation models; now, this current paper extends the findings of the local models in a regionalisation effort to the entire surface catchment area of 103 km$^2$. By contrast, the aim of this study was the generation of spatially distributed, specific recharge coefficients for every litho-stratigraphic formation in Wadi Natuf through regionalisation, i.e. the attribution and extrapolation of the recharge coefficients, modelled before at point-scale. The work was based on the understanding of dominant physical parameters and processes and carried out in two consecutive steps: In a first step, a new recharge classification framework was set up for this largely ungauged basin, based on field observations, as well as conceptualisation and classification. Relevant physical features were identified and attributed to three different groups (geology, soil and LU/LC) and within each group, different recharge classes were differentiated. In a second step, the quantification and regionalisation were carried out as an extrapolation along the above grouping and classification scheme, developed for Wadi Natuf and the WAB. Hereby, we used the location-specific recharge coefficients (RC) that were derived from the soil moisture models in Messerschmid *et al.* (2020).

The development of empirical understanding of how recharge (with empirical coefficients) takes place in deep karst formations is innovative by its own. For the first time, we were able to develop a fully distributed recharge model in a hitherto ungauged basin of deep karstic aquifers (such as Wadi Natuf in the WAB). In addition, the novelty of our approach consisted in a new combination of existing techniques that are based on observable processes, parameters and signatures. The assessment adheres to the goal of parsimony and integrates inductive and deductive steps. And by being firmly grounded in empirical surface and surface-near observations, this new approach can be transferred and applied to other, hitherto ungauged basins in order to advance the crucial but challenging task of a realistic representation of distributed recharge.

# 2   Study area

The 103 km$^2$ large catchment of Wadi Natuf extends on the western flanks of the West Bank from the Mountain crest in the east towards 1949 Armistice Line ('Green Line') in the western foothills. Much of its topography is characterized by undulating hills with deeply incised ephemeral rivers (Wadis). The catchment exhibits a pronounced spatial variability of climatic drivers (precipitation, evaporation), land use and land cover features (LU/LC), soil thickness and not least, rock lithology of the different geological (litho-stratigraphical) formations (see Fig. 2a in Messerschmid *et al.*, 2020).

## 2.1 Geology and hydrogeology

One of the reasons for choosing Wadi Natuf as an exemplary sub-catchment on the recharge zone of the Western Aquifer Basin (WAB), besides field accessibility, was the unrivalled litho-stratigraphic diversity, reaching from the deepest outcropping, Aptian formations, all the way up to the top cover series of impermeable chalks from Senonian (and Lower Tertiary) age. All formations of the WAB are covered in this study (Fig. 2b in Messerschmid

*et al.*, 2020). Together, the aquifer formations cover around two thirds (64.4 %) of the outcrop areas in Wadi Natuf; they are entirely carbonatic and in most parts strongly karstified. The other third of the area consists of outcrops of less permeable and fully impermeable formations.

According to the old, conventional view – valid on the regional scale – the regional Upper and Lower Aquifers are divided by some 100 to 150 m thick marly, chalky and carbonatic series of a so-called 'Middle Aquitard' or Yatta formation (Bartov *et al.,* 1981; SUSMAQ, 2002; Messerschmid *et al.,* 2003a, 2003b; ESCWA–BGR, 2013). The regional geology is indicated in the land use and geology map, Fig. 2 (for a detailed geological map, compare with Messerschmid *et al.*, 2020, Fig. 2b). However, closer scrutiny reveals that this regional 'Middle Aquitard' can be further subdivided. The top forms an aquitard or even aquiclude section of impermeable yellow soft marl (upper Yatta, u-Yat). By contrast, the main (lower) part of this 'regional aquitard' is more carbonatic and in parts karstified, however complemented by smaller portions of chalk, marl and chert. These somewhat marly and chalky limestones and dolomites of lower Yatta formation (l-Yat) thus form intermediate perched aquifer horizons that drain through small local springs.

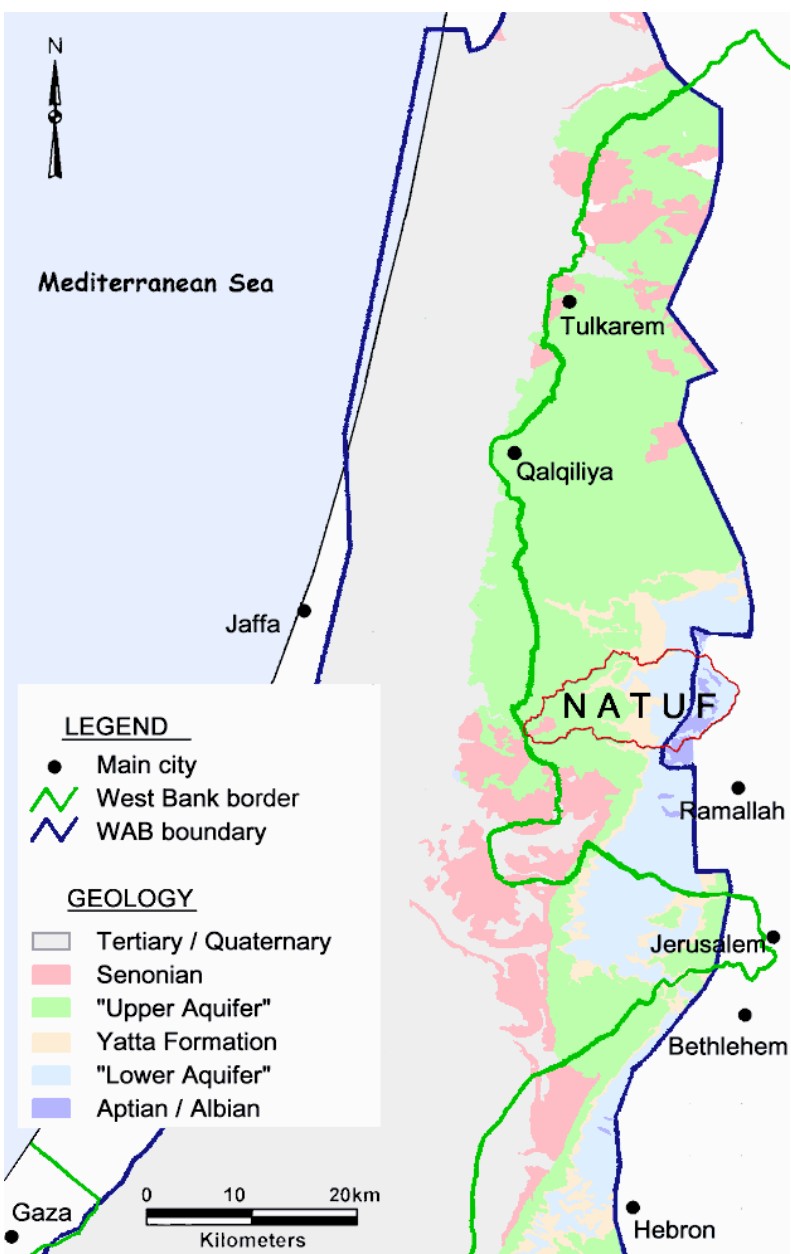

**Figure 1.** Overview of regional aquifer outcrops in Wadi Natuf and the WAB; modified after Messerschmid *et al.* (2003a).

Also, the regional 'Lower Aquifer' (LBK & UBK) can be differentiated on the local scale into more aquiferous
and more permeable parts (Table 1). Its top is formed by the conspicuous cliff-forming and very permeable reefal
limestone of upper UBK (u-UBK), that also acts as a leaky perched aquifer on the local scale (such as in Wadi
Zarqa). By contrast, the lower UBK formation (l-UBK) mostly consists of banked, often chalky dolomites (again
with intercalations of marl and chert) with a relatively poor aquifer potential. Its top however was found to be more
carbonatic but underlain by a twin marl band (Fig. 3c), which hydraulically separates the top from the main, lower
part of l-UBK and above which local contact springs align. This top of l-UBK acts as a third local and isolated
perched aquifer horizon.
By contrast, the regional 'Upper Aquifer' is void of both, perched aquifers and springs, despite the fact that it too
contains formations with thin marl intercalations of reduced permeability, such as the colourful plated limestone
series of lower Betlehem formation (l-Bet), the outcrops of which are often covered by small forests. This is due
to the presence of the thin marl intercalations which promote the development of thicker soils here (e.g. the forested
hilltop in Fig. 3a). It can thus be summarized that almost the entire Upper Aquifer and most of the Lower Aquifer
outcrops in the recharge area are void of springs.
Only the intermediate aquifers of the central study area show land forms of deeply incised erosional Wadis, which
often completely isolate the small local and often perched aquifer reservoirs on individual hills or hill groups. They
drain through over 100 hundred small and very small but perennial local contact springs (Fetter, 1994) with
individual spring flow between zero and a maximum of 1.7 l/s (Messerschmid *et al.,* 2003a, 2003b).
These isolated perched hilltop aquifers of central Wadi Natuf stand in contrast to the thick regional aquifers and
therefore only incompletely contribute to the deep regional groundwater recharge of the two regional storage and
flow systems. Together, the formations of the three isolated perched aquifer systems cover 13 % of the catchment.
The outcrop areas of all formations, as well as the differences between the local and the regional hydrostratigraphy
form one focus of the present study (see Table 1).

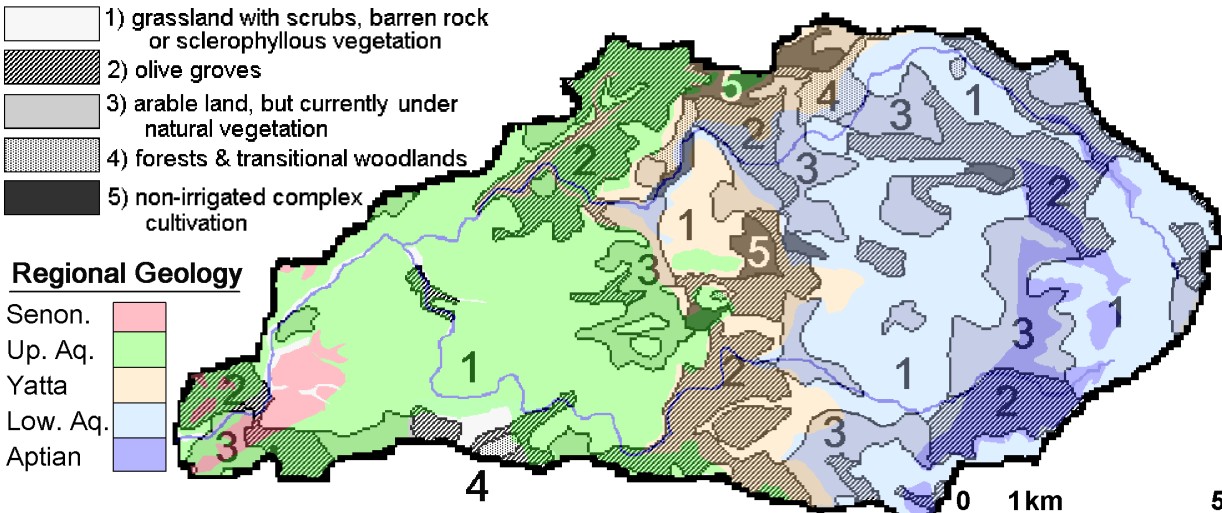

**Figure 2**. Wadi Natuf Land Use and Land Cover (LU/LC) map and regional hydrostratigraphy in shaded colours
(modified after LRC, 2004 and Messerschmid *et al.*, 2003a and ARIJ, 2012)**.**
Note: The land cover types 4 and 5 are almost completely restricted to Yatta formation outcrops (and in some places parts of
the UBK formations). Type 3 is typically found over outcrops of the regional Lower Aquifer. The Upper Aquifer outcrop area
is dominated by type 1 (grassland and barren rock). Olives (type 2) can be found in all areas, but are grown on tended terraces
mostly in the steep slopes of the Lower Aquifer outcrops in upper Wadi Natuf.

**Table 1.** Outcrop (recharge) area, average precipitation and formation names in Wadi Natuf – regional and local refined hydrostratigraphies

| Age | Area (km²) | Precipitation (mcm/a) | Formation (symbol) | Local stratigraphy, aquifer potential | Regional Stratigraphy |
|---|---|---|---|---|---|
| Recent | 1.53 | 0.85 | Alluvial (All) | (minor) | Top |
| Senonian | 2.38 | 1.31 | Senonian (Sen) | – | Aquiclude |
| Turonian | 9.24 | 5.07 | Jerusalem (Jer) | major | **UPPER AQUIFER (UA)** |
| Upper Cenomanian | 7.65 | 4.26 | u-Betlehem (u-Bet) | good | |
| | 9.77 | 5.58 | l-Betlehem (l-Bet) | poor | |
| | 10.06 | 5.77 | Hebron (Heb) | major | |
| Lower Cenomanian | 4.93 | 2.92 | u-Yatta (u-Yat) | – | Middle Aquitard |
| | 10.18 | 6.14 | l-Yatta (l-Yat) | local * | |
| Upper Albian | 2.44 | 1.50 | u-Upper Beit Kahil (u-UBK) | good * | **LOWER AQUIFER (LA)** |
| | 8.44 | 5.26 | l-Upper Beit Kahil (l-UBK) | local (at top) * / poor (at bottom) | |
| | 13.16 | 8.21 | u-Lower Beit Kahil (u-LBK) | major | |
| | 16.4 | 10.23 | l-Lower Beit Kahil (l-LBK) | major | |
| Lower Albian | 4.56 | 2.80 | Qatannah (Qat) | – | Bottom Aquiclude |
| | 1.82 | 1.12 | Ein Qiniya (EQ) | good (local) | |
| Aptian | 0.06 | 0.04 | Tammoun (Tam) | – | |
| **SUM** | **102.6** | **61.1** | | | |

Note: The area of formation outcrop here is equated with the area for infiltration (recharge). Precipitation here is expressed as average annual amount of area precipitation over the respective formation outcrops and calculated with rainfall of the respective sub-catchments within Wadi Natuf. Ein Qiniya formation is a local aquifer, which however does not belong to any of the regional aquifer units or basins; its recharge potential does not form part of the water balance calculations for the WAB. * perched leaky aquifers with dashed line at bottom; Source: this study.

## 2.2 Physical landscape features

Less than 5% of the rural Wadi Natuf landscape are built-up (Messerschmid, 2014). Its typical land forms (Fig. 2) range from rock outcrops and terraces with olives, over grass- and shrublands, arable but currently uncultivated lands, mixed vegetation and transitional woodlands to agricultural plains and forests (Messerschmid, 2014; LRC, 2004). All land forms in Wadi Natuf are closely related to the underlying geology (Fig. 3). The soft marl of u-Yat usually forms an eroded step in the landscape that can develop into small inland plains with cultivated agricultural fields. By contrast, the mixed intercalations of marly, chalky and limey rocks of l-Yat form natural steps and terraces in the landscape, often with a bushy landscape, partly also with trees. The regional aquitard of u-Yat is overlain by the strongly karstified massively bedded limestone of Hebron formation (Heb), which often restricts soil development to small pockets in an otherwise sparsely vegetated karren-field landscape. This karstic formation with an excellent recharge potential (and very low runoff generation, see Messerschmid *et al.*, 2017), in turn is overlain by the already mentioned soft, plated limestone with thin marl intercalations of l-Bet, which not only erodes differently but also allows the formation of thicker soils; Figure 3a shows l-Bet at the top of the hill, conspicuously covered by a little forest and with a sharp boundary to the LU/LC type of the underlying karstic Hebron formation.

Typically, in Wadi Natuf, this distribution of LU/LC follows the formation outcrops (geology) with great accuracy, discernible even from aerial photographs. Also soil thickness was measured and found to strongly correlate with lithology and land forms (LU/LC) as discussed in the first part of this series (see Table D1 in Messerschmid *et al.*, 2020). This recurrent field finding of strict correlation between the three groups of physical features – LU/LC, soil thickness and geology – forms the basis of the classification framework in Wadi Natuf (see sections 3.2, 3.3, 5.1), since it allows categorization of key elements of recharge and the attribution of lithological and hydro-stratigraphical characteristics with the aquifer and recharging potential of the different formations.

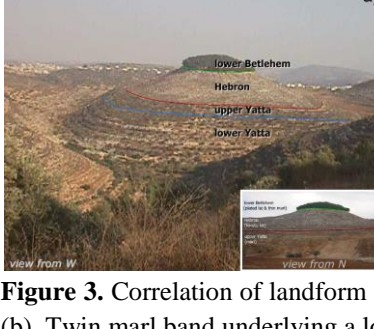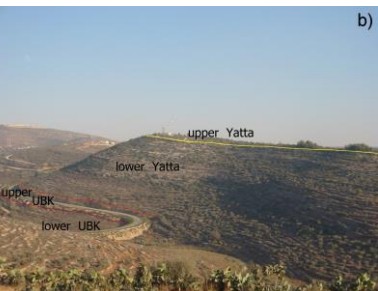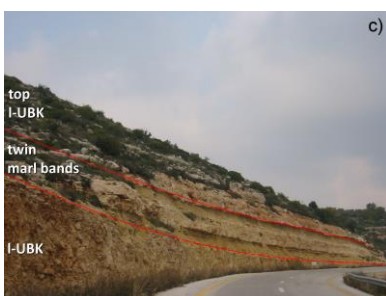

**Figure 3.** Correlation of landform and lithology. Nabi Ghayth hill, west of Beitillu (a); Nabi Aneer spring group (b). Twin marl band underlying a local perched aquifer (c).

Note: The karstic limestone of Hebron formation forms outcrops with thin soil cover, bare rock or karren fields and tends to erode into steeper slopes above the soft, mostly eroded upper Yatta formation – the only true aquiclude within the Westbank Group (with levelled agricultural plains in the inlet photo in Fig. 3a. By contrast, the top of the hill is formed by lower Betlehem formation; a thinly plated coloured limestone ensemble with fine marl interbedding that lacks karstification and promotes soil development and natural vegetation. Figure 3c shows the twin marl band, underlying and confining Beitillu, Harat Al-Wad spring group (of Top l-UBK formation).

## 3 Methodology

The regionalisation of this study employs two consecutive procedures. Step a): Identification and parameterization of physical features and their classification in a conceptual response matrix, attributed to classes of hydrological impacts (Fig. 4, rows 1 and 2). Step b): Extrapolation and regionalisation of the quantitative model results from Messerschmid *et al.* (2020) within a classification framework (row 3).

### 3.1 Physical features

The classification of distributed physical landscape features and their parameters stands at the heart of this study. Mapping, detection, interpretation and where possible, quantification of their parameters was carried out over a period of more than ten years and over 200 field visits to gain local knowledge on specific field conditions.

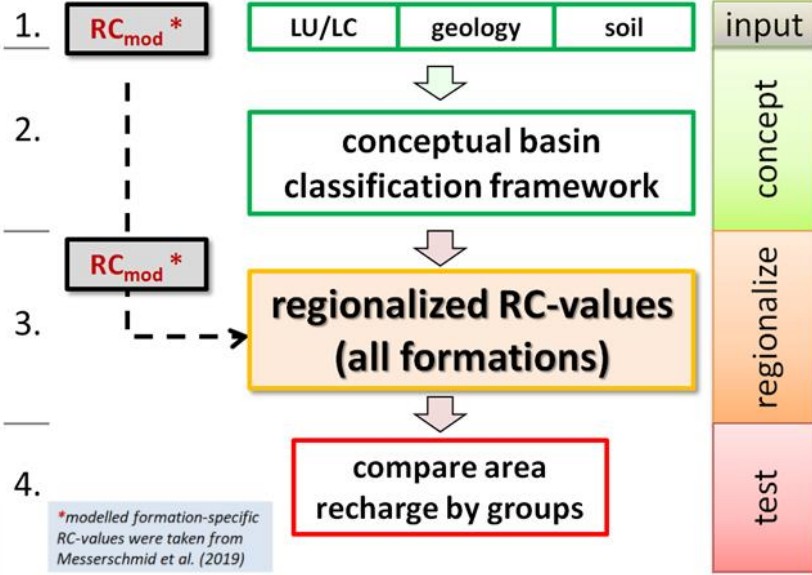

**Figure 4.** Conceptual flow diagram of work steps

First row: field observations on land forms, geology and soil, together with key date campaign on spring flow measurements; second row: setting up a conceptual classification framework; third row: introducing formation-specific RC-values (from Messerschmid *et al.*, 2020) and regionalisation of RC-values for the entire catchment (all formations and all three groups); fourth row: area recharge calculation and comparison of results for the different groups.

First, existing geological maps in the scale 1:50,000 (GSI, 2000; 2008; Rofe & Raffety, 1963) were corrected, complemented and refined by extensive field mapping and remote sensing (stereoscopic aerial photographs) with the target to detect, describe and interpret the lithological rock content, (mineralogical composition, texture, grain distribution), the degree of crystallisation and structural features like folding, faulting, cleavage, jointing, as well as

primary porosity and karstic features. These karstic features encompass epi-karst surface features (karren and schratten landscape), karstic solution holes (especially in the l-LBK formation, aka "Swiss cheese" formation and in parts of the Hebron formation), well-known caves and karstic channels in the underground; wide, karstified fractures (incl. their width and prevalence). The study also built on earlier small-scale fracture trace and lineament mapping in the Ramallah district (Aliewi and Messerschmid, 1998). In addition, there are indirect indicators of sub-surface karst (such as rapid interflow emerging at "bleeding hills" and travertine crusts). Another focus was the refinement of local hydrostratigraphy, in particular with respect to the spring-feeding formations (Messerschmid *et al.*, 2003b; Dafny *et al.*, 2009) and their catchment areas. Of particular interest were not only the spatial pattern and distribution of such features, but especially the comparison of these geological features with the features and distribution of the other groups, i.e. soil and LU/LC. This enabled us to assign particular, spatially distributed geological characteristics to each of the different formations. This study first re-examined the distribution of landscape features with respect to their recharge potential and the exact delineation of the outcrop and recharge areas of the different aquifer and aquitard formations.

The second part of field mapping and investigations targeted the soils in Wadi Natuf. Lab tests found silty to clayey residual soils (terra rossa), which are typical for Mediterranean carbonate environments (see also Messerschmid *et al.*, 2020). The main aim of this sub-study was to investigate soil thickness and its distribution over the area. As already mentioned a conspicuous spatial pattern emerged, namely that typical soil thicknesses formed over different formations (see Fig. 3a, 3b). Appendix D in Messerschmid *et al.* (2020,) presents these results in a soil thickness matrix, where the distribution of soil depth was documented for different LU/LC-types and different lithostratigraphic units.

Thirdly, and similar to geology and soil thickness, land use and land cover characteristics, such as relief, natural vegetation and its alteration by human land use (section 2), can be interpreted as indicators of different hydrological processes that determine recharge. Whereas the differences in landscape units with respect to their runoff potential were discussed in Messerschmid (2014); Messerschmid *et al.* (2018), this study aimed at creating a simplified but realistic categorization of physical, recharge-controlling landscape features and their spatial distribution along the lines of outcropping formations.

**3.2 Conceptual basin classification framework and regionalisation**

Conceptually, as already mentioned, the regionalisation in this study comprises of two main steps (rows 2, 3 in Fig. 4), i.e. the creation of a basin classification framework and the attribution of the model results of Messerschmid *et al.*, (2020) to this framework by extrapolation and regionalisation, which will be further specified in the following. Based on the PUB-understanding that physical characteristics control hydrological processes and thus (hydrological) function follows (physical) form, a conceptual framework was set up, as shown in Table 2. The physical features were divided into three groups, such as LU/LC, soil and geology (columns in Table 2), and within each group separately, the different landscape units were divided into distinct classes of recharge potential (lines in Table 2), based on the available geological literature and our extensive field investigations. Then, each lithostratigraphic formation (numbered a, b, c, etc. in the schematic Table 2) was attributed to a distinct recharge class (from low to high in Table 2; as roman numbers I – V in Table 3). As a result, we obtained three independent sets of differently ordered litho-stratigraphic formations, ranked by their recharge potential. This separation allowed us to examine the result of attributed recharge classes separately for each group in order to gain a more realistic picture, to examine the differences in outcomes and to avoid over-simplification in line with PUB-recommendations (section 1.2). Again, this procedure was based on the findings of section 2, namely that such a correlation between the three groups of physical features was clearly discernible in field explorations in Wadi Natuf. It should be noted here that whereas soil thickness was a quantifiable parameter, other physical parameters such as LU/LC (types of natural vegetation or land use) and geology were not. So, LU/LC & geology features were ranked according to their recharge potential and correlated with soil thickness (see: ranked classes of recharge potential from low to high, as in Table 2). This was based on a soil thickness correlation matrix in which representative typical soil depths could be attributed to the different formations and land forms (this soil matrix is shown in Appendix D, Messerschmid et al., 2020).

Five different formation-specific RC-values were obtained from the monitored and modelled SM-station data (Messerschmid *et al.*, 2020), representing 5 different classes of recharge potential (between 57% and 42%, shown in bold red font in Table 4). In addition, impermeable formations were not monitored at SM stations; Instead, a sixth class of zero recharge (RC = 0%) was added for them. According to the correlation and grouping in the BCF, the different specific RC-values of the modelled formations could be attributed to other formations as well (see Table 4). Hereby, the exact RC-values found in the soil modelling were redistributed and assigned to different formations under LU/LC and geology groups – shown schematically in Table 2, conceptually in Table 3 and quantitatively in Table 4. By this step, all existing formations in Wadi Natuf were assigned specific RC-values of annual recharge (ranging from 57% to 42% in aquifers and down to zero recharge in the aquitards). It should be added here that for three formations, an additional step was needed. These formations were found not to be uniform but either consist of different sub-facies at different locations of the catchment (e.g. u-Bet and l-Bet) or represent different lithologies in the stratigraphic column (within l-UBK formation, see photo, Fig. 3c, above). In these instances, an additional, intermediate RTC value was introduced, based on the arithmetic mean of classes II and IV (49.4% - as average between 44.7% and 54.1%).

**Table 2.** Schematic conceptual basin classification framework

| | | Groups of physical features | | | | | |
|---|---|---|---|---|---|---|---|
| | | LU/LC | | Soil thickness | | Rock lithology | |
| | | Ftn. | Phys. features | Ftn. | Phys. features | Ftn. | Phys. features |
| High | Classes of recharge potential | a) | Rock | b) | Thin | c) | Karst |
| ↑ | | b) | Grassland | a) | Medium | b) | Limestone |
| | | c) | Forest | c) | Thick | a) | Marl |
| Low | | | | | | | |

Note that the order of formations (a, b, c), differs from group to group, thus indicating different ranking orders of formations as to their recharge potential (classes) in each group. This table is a generic example; more classes can be used.

## 4    Results

### 4.1 Basin Classification

This analysis results in a basin classification framework that categorizes different groups of recharge potential and attributes each formation to one of these classes, shown in Table 3. Each formation is attributed to different classes of recharge potential (lines) and independently for each "group" of physical features (columns). Hereby, the ranking order of some of the formations differs from group to group, based on literature and field observations as well as on conceptual considerations grounded in general physical laws, (see Fig. 2; sections 2 and 3).

### 4.2 Regionalisation and extrapolation of modelled RC-values

Using this basin classification framework, it was now possible to extrapolate the results of the parsimonious percolation model and attribute the modelled recharge coefficients to other formations (according to classes of recharge potential, Table 3). To avoid equifinality problems and increase the reliability of the approach, this attribution of RC-values was performed for each group of physical features independently. This approach rests on the assumption that the seven-year observation period fairly represents long-term variability of inter-annual rainfall (see Messerschmid *et al.,* 2020; App. E). Table 4 shows the modelled and the newly attributed and inserted average annual recharge coefficients for each group. In the table, those RC-values, which are directly taken from the model (Messerschmid *et al.,* 2020) were marked in bold font and red colour (in group 2, representing soil thickness).

**Table 3.** Conceptual basin classification framework, specific for Wadi Natuf

| | group 1 - LU/LC | | group 2 - soil | | group 3 - geology | | |
|---|---|---|---|---|---|---|---|
| | formations | features | formations | feat. | formations | features | |
| I | | | All, **Jer**, l-LBK | -- | All, Jer, l-LBK | well dev. karst (& gravel) | Increasing recharge potential → |
| II | u-UBK | cliff, mostly rock outcrops | **u-UBK**, u-LBK | - | Heb, u-UBK, u-LBK | karstified lst / dol | |
| * | | | | (…) | l-Bet, u-Bet, l-UBK | lst / dol (some marl / chalk) (*Nari for u-Bet*) | |
| III | All, Jer, Heb, u-LBK, l-LBK | olive terraces, rock outcrops | u-Bet, **Heb**, EQ | -/+ | | | |
| IV | u-Bet, l-UBK | arable but uncultivated, grass- & shrublands | **l-UBK** | +/- | | | |
| V | l-Bet, l-Yat, EQ | mixed, transit. woodlands | **l-Yat**, l-Bet | + | l-Yat, EQ | mixed lst + marl | |
| - | *(as Gr.2)* | agric. plains, forests | Sen, u-Yat, Apt | ++ | *(as Gr.2)* | marl (chalk) | |

*Note*: Left column: classes of measured recharge potential (I – V); middle columns: groups of phys. features (1-3); formation names as in Table 1; soil thickness increases from thin (--) to thick (++). The formations shown in bold type were the ones monitored, measured and modelled. The grouping and class distribution was based on field work and literature, e.g. SUSMAQ (2002), LRC (2004), GSI (2001), Keshet and Mimran (1993), Messerschmid (2014) and Messerschmid *et al.* (2018). Aquitards, i.e. impermeable formations, where recharge is assumed zero, were not measured in SM-stations (bottom line of Tab. 3).

Regarding the class marked with asterisk *, this formation was not measured in SM-stations. Instead, for group 3 (Geology), the average of RC for classes II and IV was taken (as 49.4 %), because these formations appear in two facies types, which are more and less permeable, respectively.

Table 4 lists the 15 different outcropping formations in Wadi Natuf in chronological order from the youngest, alluvial series and impermeable Senonian chalks down to the oldest, also impermeable lower Albian – upper Aptian Tammoun shales formation (Messerschmid, 2003a). In between, there are two aquitardal series (Qat, u-Yat) and ten more or less aquiferous formations, almost all of which are partially composed of carbonates (including the unconsolidated carbonate gravels forming the shallow alluvial in the Wadis). However, the recharge coefficients (as fraction of rainfall) of the aquifers deviate by over 15 % between the most susceptible karstic limestones with an RC > 57 % and the more aquitardal series, containing some degree of marl and chalk, be it as discrete thin beds (l-Bet) or as marly and chalky limestones (l-Yat) with an RC of almost 42 %. These high recharge rates are partly due to the much reduced (in fact negligible) rates of runoff generation measured in Wadi Natuf. But more importantly, they are a result of the overall quite modest amounts of actual evapotranspiration, caused by the Mediterranean climate with a rather short but very wet winter season and a prolonged rain-free summer season, in which the dried-up soils cannot offer any amounts of water to direct soil evaporation or plant transpiration, undergoing a kind of summer dormancy.

As described before and applied in the modelling code of the SM-saturation excess and percolation model, the rate of groundwater percolation (here equated with recharge) from the soil into the aquifer bedrocks is directly related to the thickness of the soil. In other words, our model and hence also this table is based on the observation and assumption that thicker soils permit less recharge than thin soil covers. Consequently, the highest RC-values are all found in formations with very thin soils and larger portions of rock outcrops (around soil pockets), such as the Turonian limestones of Jerusalem formation at the top of the regional Upper Aquifer and as the bottom of the regional Lower Aquifer, the lower LBK formation (l-LBK), both of which display highly karstified and massively bedded limestone series with strong features of epikarst in the outcrop (SUSMAQ, 2002). These formations also show the highest recharge coefficients in the physical landscape feature group 3 (geology), due to the aforementioned lithological features. However, under the third group (LU/LC), these formations rank lower than the maximum RC-values (instead, the cliff-forming u-UBK formation reaches the maximum here). This is due to the fact that, from a land use and land cover point of view, these two formations had to be grouped into class II of recharge potential (see Table 3), because here, besides the extended grass- and scrub lands, olive groves dominate

on the cultivated terraces of l-LBK and on the plains of Jerusalem formation (see LU/LC map, Fig. 2). The other,
un-modelled aquifer formations (u-Bet, Hebron, u-LBK and the stratigraphically deep formation Ein Qiniya) are
attributed with intermediate RC-values (with 0% for aquitards and 57 % for the highest potential), according to
their class of recharge potential (Table 3).
**Table 4.** **Extrapolated recharge coefficients per group**

| Formation | Area km² | Precipitation mcm/a | Precipitation mm/a | 1. LC/LU Recharge RC (%) | 1. LC/LU Recharge mm/a | 2. Soil Recharge RC (%) | 2. Soil Recharge mm/a | 3. Geology Recharge RC (%) | 3. Geology Recharge mm/a |
|---|---|---|---|---|---|---|---|---|---|
| Alluvial | 1.53 | 0.85 | 553 | 45.3% | 250 | 57.3% | 317 | 57.3% | 317 |
| Senonian | 2.38 | 1.31 | 552 | 0.0% | 0 | 0.0% | 0 | 0.0% | 0 |
| Jerusalem | 9.24 | 5.07 | 549 | 45.3% | 249 | **57.3%** | 315 | 57.3% | 315 |
| u-Betlehem | 7.65 | 4.26 | 557 | 44.7% | 249 | 45.3% | 252 | 49.4% | 275 |
| l-Betlehem | 9.77 | 5.58 | 571 | 41.8% | 239 | 41.8% | 239 | 49.4% | 282 |
| Hebron | 10.06 | 5.77 | 574 | 45.3% | 260 | **45.3%** | 260 | 54.1% | 311 |
| u-Yatta | 4.93 | 2.92 | 592 | 0.0% | 0 | 0.0% | 0 | 0.0% | 0 |
| l-Yatta | 10.18 | 6.14 | 603 | 41.8% | 252 | **41.8%** | 252 | 41.8% | 252 |
| u-UBK | 2.44 | 1.50 | 615 | 54.1% | 333 | **54.1%** | 333 | 54.1% | 333 |
| l-UBK | 8.44 | 5.26 | 623 | 44.7% | 279 | **44.7%** | 279 | 49.4% | 308 |
| u-LBK | 13.16 | 8.21 | 624 | 45.3% | 283 | 54.1% | 338 | 54.1% | 338 |
| l-LBK | 16.4 | 10.23 | 624 | 45.3% | 283 | 57.3% | 358 | 57.3% | 358 |
| Qatannah | 4.56 | 2.80 | 613 | 0.0% | 0 | 0.0% | 0 | 0.0% | 0 |
| Ein Qiniya | 1.82 | 1.12 | 613 | 41.8% | 256 | 45.3% | 278 | 41.8% | 256 |
| Tammoun | 0.06 | 0.04 | 614 | 0.0% | 0 | 0.0% | 0 | 0.0% | 0 |
| **SUM / avg.** | **102.6** | **61.1** | *595.3* | **39.5%** | **235** | **43.8%** | **261** | **46.0%** | **274** |

Note: The modelled RC-values, taken from Table 2 in Messerschmid *et al.* (2020), are indicated in red and bold fonts under
the second group (soil conditions). Aquitards void of recharge are shaded grey.
Note again that Wadi Natuf comprises of a main part belonging to the WAB, a smaller Eastern portion (in the
mountains) belonging to the groundwater catchment of the EAB and reduced outcrop areas, older than and
stratigraphically below the bottom formations of the regional Lower Aquifer in both, WAB and EAB.  Table 5
documents the total recharge in Wadi Natuf (as well as that of the WAB portion only, in brackets and blue colour).
The resulting overall area recharge coefficient for the entirety of Wadi Natuf ranges between 39.4 % and 46.1 %,
slightly higher for the WAB portion (44.2 % as mean value of the three groups). As can be noted, despite the
independent approaches and individual RC-attribution for each group, the final results of average area recharge
within the WAB portion match rather closely for each calculation, with 24.1, 26.8 and 28.1 mcm/a, respectively,
or in other words, with a deviation of total distributed recharge by less than 10 percent.
Figure 5 shows the overall catchment recharge as results of the three independent runs of regionalisation for each
of three types of landscape characteristics (geology, soil and LU/LC). The values from Table 4 were applied here
and mapped as visualisation. The overall results of the three runs match closely. The ranking of formations
according to LU/LC resulted in the lowest overall recharge. The regionalisation according to geology shows the
highest values. The values of the soil-based group (middle column, marked bold in Table 4) take an intermediate
position, close to the arithmetic mean of the three groups. A more detailed translation of the recharge values for
different stations into area and aquifer recharge rates is documented in Table A1.
**Table 5.** Annual average recharge in Wadi Natuf for different groups of landscape features – (WAB only)

| Scenario | Unit | Recharge – all Natuf ( WAB ) Group 1 landform-based | Recharge – all Natuf ( WAB ) Group 2 soil-based | Recharge – all Natuf ( WAB ) Group 3 lithology-based |
|---|---|---|---|---|
| **Recharge** | ( mcm/a) | **24.1 (20.6)** | **26.8 (22.6)** | **28.1 (23.9)** |
| **Catchment area** | ( km²) | | **102.6 (85.5)** | |
| **Average precipitation** | ( mm/a ) | | **595** | |
| **Annual recharge rate** | ( l/m²/a ) | 0.23 (0.24) | **0.26 (0.26)** | 0.27 (0.28) |
|  | ( mm/a ) | 234.8 (241.4) | **261.4 (264.2)** | 274.1 (279.8) |
| **Recharge coefficient** | ( % ) | 39.4 % (40.8 %) | **43.8 % (44.6 %)** | 46.1 % (47.3 %) |

29 Note: mcm/a = million cubic-metres per year, the blue numbers refer only to the WAB-portion with Wadi Natuf.

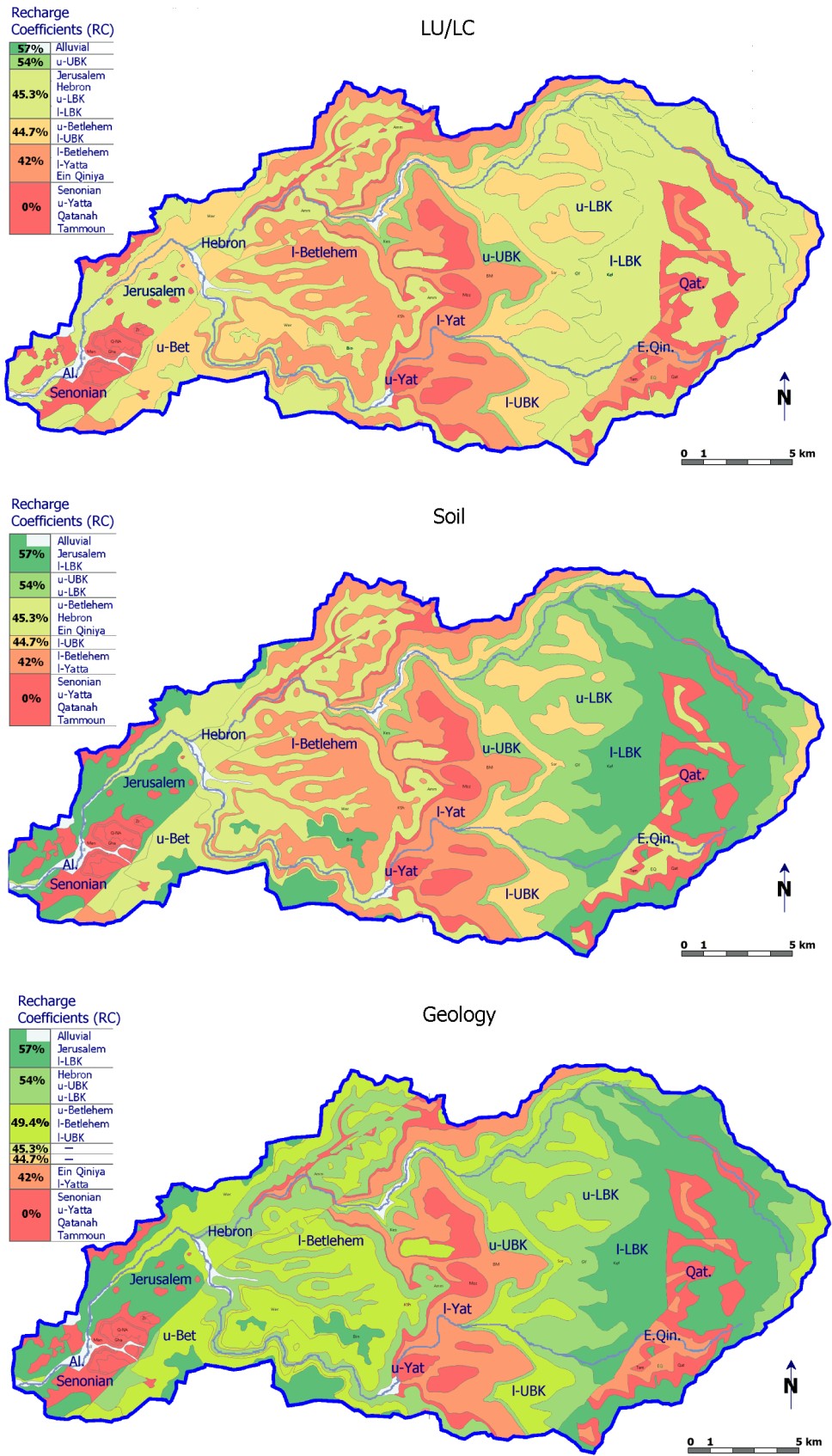

**Figure 5.** Recharge map of Wadi Natuf – Recharge Coefficients of the different formations according to the different groups of landscape characteristics (LU/LC, soil and geology). The RC-values are shown in % of annual precipitation.

## 5 Discussion

### 5.1 General approach of process representation

PUB research had previously suggested new ways to describe and estimate distributed basin responses, but mostly focussed on runoff rather than on recharge and its spatial distribution.

Savenije (2010) suggested assigning individual hydrological processes and distinct hydrological functions (e.g. runoff) to different landscape units by dissecting catchments in a semi-distributed way and according to a hydrologically meaningful landscape classification metric. Batelaan & de Smedt (2001) accounted for spatial variation of physical features using a water budget of rain, evapotranspiration and runoff. In Batelaan and de Smedt (2007) long-term recharge largely depended on soil and LU/LC differences (with parameters based on literature values). Aish, Batelaan and de Smedt (2010) could not only draw on physical features but also on hydrological basin response knowledge (water levels) in their water balance model of the Gaza Strip (similarly Tillman *et al.*, 2015). Several authors used dimensionless numbers of 'similarity patterns' to relate physical form to hydrological impact in basin-wide transfer functions (Berne *et al.*, 2005; Woods, 2003 and Radulović *et al.*, 2011). Other authors calibrated parameters of the transfer functions such as soil properties (Ali *et al.*, 2012) or LU/LC, soil and geology (Götzinger, 2006) or land forms such as depressions (Baalousha *et al.*, 2018). Simple soil water models at the basin scale for daily recharge estimates in temperate climates were used by Dripps *et al.* (2007) and by Finch (2001) as responses to land cover changes.

This study went a step further than the existing literature and employed new methods by combining deductive (conceptual) and inductive (empirical) approaches to determine spatial variations in groundwater recharge, based on qualitative (dimensionless) and on measured quantitative basin observations alike. Our distribution into distinct classes of recharge potential, we would like to stress here again, was an act of attribution and deduction; it was however, firmly grounded in general physical laws, such as permeability of different lithologies or different forms of land use. Only the second group of soil moisture was empirically quantified by repeated field measurements in the form of soil depth probing. The results of this empirical survey of soil depth distribution for different hydrostratigraphical units are documented in the soil depth matrix in Messerschmid *et al.* (2020), Table D1.

Our three independent runs of conceptual analysis and attribution resulted in a close range of total WAB recharge (24, 26 and 28 mcm/a). This suggests that our transfer procedures delivered a robust and realistic representation of the processes at hand (which we prioritized over allegedly "exact" but less reliable results).

In our study, we tried to avoid problems of multicollinearity and equifinality by testing three conceptual approaches individually and separately in different groups according to physical basin form (grounded in empirical observation). It should be noted here that this approach was based on general knowledge and understanding of processes that can be observed worldwide; for example, high recharge potential can be attributed to areas with barren rock but also to terraces with tended olive groves, where runoff is inhibited by stone walls, where soils are relatively thin and farmers plough and remove weeds twice a year, which in turn reduces plant transpiration and thus slows down the loss of soil moisture. On the other hand, forests and agricultural plains with thick accumulated soils are known to reduce the infiltration, percolation and hence recharge potential. The same is true for different lithologies of receiving bedrock (like carbonatic, argillaceous and arenitic sediments). Although applicable worldwide in principle, our approach of separately accounting for three land feature groups signals a departure from many of the existing studies in other areas, which probably over-simplified matters by combining and subsuming all types of typical landscape features in one group, which then were split into different classes of basin responses.

As already mentioned, and by contrast to most earlier studies in the WAB, the focus of our approach was clearly the spatial, not temporal distribution and variability of recharge. This work was based on two assumptions: a) that the seven-year rain period of the SM-percolation model is a fair representation of long-term averages of both inter-annual and seasonal distribution of precipitation (see Messerschmid *et al.*, 2020) and b) that each of the selected SM stations is representative of the entire formation. Here we draw on the above results for the spatial distribution

of physical features (Table 2) and soil depth (Table D1 in Messerschmid *et al.*, 2020) of the respective formations. The results of these measurements and analysis confirmed the well-known fact that the temporal distribution of precipitation events strongly affects the percolation rates.

As the main aim of our research, we thereby obtained a detailed differentiation of the spatial distribution of recharge with formation-specific recharge coefficients for all formations in Wadi Natuf, which is a representative catchment for the recharge area of the WAB. The results of our three-way conceptual analysis and attribution seemed to suggest that indeed, slightly different results of overall recharge rates follow from the three approaches. However, the relative closeness of the three results, e.g. the total WAB recharge in Wadi Natuf of 24, 26 and 28 mcm/a, respectively, did suggest that each of the three independent transfer procedures between basin form and response was a realistic representation of the processes at hand. In other words, instead of producing an apparently precise figure for groundwater recharge, our analysis resulted in a less "exact" but more robust realistic and nonetheless close range of recharge quantifications.

## 5.2 Limitations and Caveats

To begin with, the results of the SM-models, the RC-values of the different formations as input data for our basin classification framework (BCR), are taken as correct and reliable. A discussion of the limitations and caveats of the results and methods can be found under Messerschmid *et al.* (2020). However, the process of setting up a BCF and attributing different classes of recharge potential (RP) to the different physical features (under the 3 groups selected) is a deductive step, which relies on the translation of qualitative observations in the field into quantitative classes of RP. Therefore, the exact classes under the here developed BCF, although based on and rooted in universal physical laws and well-established evaluations, could be somewhat imprecise and incorrect. Some classes could have been selected wrongly and could under- or overestimate certain factors (features) for the decision. This is why we found it imperative to establish three independent runs of classification for the three different groups of indicators, which allows us to weigh and compare and thus evaluate the reliability of the BCF.

Another theoretical possibility is that some processes, although present in the field, were not detected and included in the set-up of the BCF. However, the approach of this study used the most commonly known principle groups of physical landscape criteria quoted in most of the literature (see Ch. 1, Introduction). Therefore it can be stated with confidence, that the processes covered by our selection of classification criteria belong to the most important, principle processes of GWR and it is rather unlikely that a major process was overlooked.

The possibility of overlooking a minor process is always and necessarily a by-product of such simplification. Hence, such simplification, a major characteristic of our approach is not only a strength but also a (relative) weakness. However, it should be repeated here once again that the need for simplification of the host of processes at work in groundwater recharge is strongly recommended and explicitly highlighted by the existing PUB-literature (see Ch.1, Introduction). In addition, the overall results were also weighed against and compared with similar results from other catchments, especially such in in the WAB and its environs.

Lastly and although the correlation between the three groups of observable features was clearly observed and investigated in depth within Wadi Natuf, it may be absent in other catchments. This then would pose a limitation to the applicability of the approach chosen. However, in such a case, other correlations can and should exist; they should be studied and detected individually for each other basin, but otherwise following the same approach as designed for this study.

## 5.3 Annual RC – overall basin RC – compared with other studies

As presented already, the individual recharge coefficients for the different formations cropping out in Wadi Natuf lie between a minimum of 0% (non-recharging formations) and a maximum of 57%. For the WAB portion of Wadi Natuf, the total average recharge for each group was found at 20.6, 22.6 and 23.9 mcm/a, respectively. (This is equivalent to a WAB recharge coefficient of 40.8%, 44.6 % and 47.3 %, respectively, within Wadi Natuf.) These

overall recharge values fall well into the range, usually quoted for the WAB (see Table A1). Also compare with the detailed table in Appendix H of Messerschmid *et al.* (2020) that lists the regional and other reported recharge coefficients, both for annual and event-based calculations and together with the methods applied therein. Weiss and Gvirtzman (2007) reported maximum recharge for one outstanding year (1988) as 91 % of annual rainfall at the small Ein Al-Harrasheh catchment on the SE edge of Wadi Natuf (Table H1). Allocca *et al.* (2014) found in the Apennine that for single events, up to 97 % of event precipitation may percolate and arrive as recharge at the groundwater table. Rosenzweig (1972) reported that for pasture and grassland at Mt. Carmel Basin, land form-specific recharge can amount to 60 % of annual precipitation. Allocca *et al.* (2014) quoted average annual RC-values ("effective infiltration") from other countries (Hungary, Greece, Spain, France and Croatia) to range from 35% to 76 % and of 27 % for Tennessee (dolomites) and found recharge coefficients of 50 – 79 % in their own study in the southern Apennines. Martos-Rosillo *et al.* (2015) present a review of groundwater recharge studies in Spain. They found spatial variations due to: "the degree of surface karstification and the development of the vegetal cover–soil–epikarst system in the carbonate aquifers". "The recharge may range anywhere from 7 to 720 mm/year. The mean coefficient infiltration or recharge rate is 38 % of the rainfall, ranging between 4 and 62 %." Our findings of a range between <40 % and >47 % of overall annual recharge coefficients lie well in the middle of reported literature (incidentally, Weiss and Gvirtzman's average RC of 47.2 % for Harrasheh sub-catchment matches exactly with our maximum area RC of 47.3 %). By contrast, RC-values determined in recent studies in the Eastern Aquifer Basin at 33 % in the upper slopes (Ries *et al.,* 2015) and 25 % in the lower slopes near the Jordan Valley (Schmidt *et al.,* 2014) ranged somewhat lower; this is according to expectations due to the more arid climatic conditions with less precipitation and higher evaporation rates.

# 6 Conclusions

This study contributes to the assessment of distributed recharge in a Mediterranean karst area with a pronounced annual rainfall pattern of two seasons (dry and wet) and with a high variability in lithostratigraphy and other related landscape features, a key topic under future climate change. In line with the findings of the PUB decade, it was possible to solidly ground our basin classification for dominant recharge processes in observations of the physical form and based on fundamental laws of physics. Although the exact combination of land features is unique the catchment at hand, its individual physical features and processes are common in many other Mediterranean catchments as well as worldwide: Relatively thin, clayey terra rossa soils covered by semi-arid to sub-humid climate vegetation; a highly variable relief with undulating hills, deeply incised Wadis and small inland plans; a pronounced seasonal precipitation; soil infiltration and runoff dominated by soil moisture saturation and storage; and last, not least, a series of well-bedded carbonates that are subject to uplifting, tilting and pronounced erosion, such as karstification. These characteristics were observed, analysed and united in a common basin classification framework (BCF). This new, intrinsic approach enabled a more precise quantification of recharge and of the areas concerned by this recharge (which can therefore be more protected for example). We found an accentuated spatial variability of percolation fluxes and a strong dependency on three main groups of physical form, namely LU/LC, soil thickness and geology. For the first time in the WAB, our study used a truly distributed approach for a great variety of different physical land forms by employing extensive direct field observations and intensive multi-seasonal measurements. To extrapolate our findings, we ran three independent sets of basin classification and grouping in classes of recharge potential as observed in our study area.

While our regionalised recharge coefficients originated from plot-scale measurements, the results matched closely with long-term observations reported in the WAB literature. The application, attribution and extrapolation of these coefficients for other, unmonitored formations reflect the ranges of recharge reported in the same region (WAB and environs) by previous studies that used lumped outflow-based basin-wide modelling (without spatial recharge differentiation).

On the side of spatial differentiation and given the lack of existing hydrological measurements, our approach followed the three-way compromise prescribed by PUB (Beven and Kirkby, 1979) between the advantages of model simplicity, the complex representation of spatial variability of hydrological basin response and the economic limitations on field parameter measurement. This was done by applying the simple cooking recipe of Sawicz *et al.*

(2011) for regionalisation in ungauged basins, namely classification (to give names), regionalisation (to transfer information) and generalization (to develop new or enhance existing theory).

Whereas our BCF for Wadi Natuf is site-specific, the general approach of using physical characteristics in poorly gauged basins can be readily applied to other catchments around the world, with only minor modifications in order to achieve meaningful predictions and a full representation of the spatial distribution of groundwater recharge even in the absence of plentiful groundwater observation points.

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

# A P P E N D I X

Table A1 below lists the detailed results of the regionalization of RC-values for individual formations (see also Table 4) and independently for each group of physical features. The table refers to the

entire catchment, including WAB, EAB and the erosion zone between the two. The arithmetic mean of the results of all three physical feature groups is indicated in the column to the right. The ranges

of recharge coefficient for individual formations lay between 57% and 0% of annual rainfall, each depending on the individual land use, geology and soil type conditions of the formation. The order

of formations in this table is listed as groups of differing aquifer potential (second column from the left), from the very permeable and productive regional aquifers reaching, in average of all three

physical feature groups to over 50% RC (strong blue) down to in average 42% RC for the weak, somewhat aquitardal local aquifers (brown fonts). The aquitards are assumed as impermeable and

contributing no recharge. The relative weight of recharge of each aquifer type group is indicated under "group fraction", indicating the contribution of each group of respective aquifer types between

almost 60% (regional aquifers) and only 10% (weak aquifers) of total recharge, summing up to 100%.

The average of total area recharge in Wadi Natuf as arithmetic mean of the three physical landscape feature groups lies at 43.1 %. It should be noted that although the regionalisation was performed

for each group of physical features independently, the differences in individual formations equal out to very similar overall recharge rates of approximately 27 $\pm$2 mcm/a (or as percentage, between

39% and 46%), as average over the seven-year measurement and modelling period.

**Table A1. Recharge of all formations and aquifer groups in all of Wadi Natuf, detailed by groups of physical features (as coefficients and annual recharge rates)**

| For-mation | Aquifer Group | Area km² | Ø P mcm/a | RC (%) LU/LC | RC (%) Soil | RC (%) Geol | Recharge (mcm/a) LU/LC | Recharge (mcm/a) Soil | Recharge (mcm/a) Geol | Group Rech. (mcm/a) LU/LC | Group Rech. (mcm/a) Soil | Group Rech. (mcm/a) Geol | Group fraction (%) LU/LC | Group fraction (%) Soil | Group fraction (%) Geol | Group RC, Natuf (%) LU/LC | Group RC, Natuf (%) Soil | Group RC, Natuf (%) Geol | Ø RC (%) Σ Natuf |
|---|---|---|---|---|---|---|---|---|---|---|---|---|---|---|---|---|---|---|---|
| Al | Alluvial | 1.5 | 0.8 | 45.3% | 57.3% | 57.3% | 0.4 | 0.5 | 0.5 | 0.4 | 0.5 | 0.5 | 1.6% | 1.8% | 1.7% | 45.3% | 57.3% | 57.3% | 53% |
| l-LBK | Strong Regional Aquifer | 16.4 | 10.2 | 45.3% | 57.3% | 57.3% | 4.6 | 5.9 | 5.9 | 13.3 | 15.8 | 16.3 | 55.1% | 59.1% | 58.1% | 45.3% | 54.0% | 55.8% | 52% |
| Jerus | | 9.3 | 5.1 | 45.3% | 57.3% | 57.3% | 2.3 | 2.9 | 2.9 | | | | | | | | | | |
| u-LBK | | 13.2 | 8.2 | 45.3% | 54.1% | 54.1% | 3.7 | 4.4 | 4.4 | | | | | | | | | | |
| Heb | | 10.1 | 5.8 | 45.3% | 45.3% | 54.1% | 2.6 | 2.6 | 3.1 | | | | | | | | | | |
| u-UBK | Inter-mediate Aquifer | 2.4 | 1.5 | 54.1% | 54.1% | 54.1% | 0.8 | 0.8 | 0.8 | 7.4 | 7.4 | 8.3 | 30.7% | 27.7% | 29.4% | 44.6% | 44.7% | 49.8% | 46% |
| l-UBK | | 8.4 | 5.3 | 44.7% | 44.7% | 49.4% | 2.4 | 2.4 | 2.6 | | | | | | | | | | |
| u-Bet | | 7.7 | 4.3 | 44.7% | 45.3% | 49.4% | 1.9 | 1.9 | 2.1 | | | | | | | | | | |
| l-Bet | | 9.8 | 5.6 | 41.8% | 41.8% | 49.4% | 2.3 | 2.3 | 2.8 | | | | | | | | | | |
| E.Q. | Weak Aquifer | 1.8 | 1.1 | 41.8% | 45.3% | 41.8% | 0.5 | 0.5 | 0.5 | 3.0 | 3.1 | 3.0 | 12.6% | 11.5% | 10.8% | 41.8% | 42.3% | 41.8% | 42% |
| l-Yat | | 10.2 | 6.1 | 41.8% | 41.8% | 41.8% | 2.6 | 2.6 | 2.6 | | | | | | | | | | |
| Senon | Aquitard | 2.4 | 1.4 | 0% | 0% | 0% | 0 | 0 | 0 | 0 | 0 | 0 | 0% | 0% | 0% | 0% | 0% | 0% | 0% |
| Qat | | 4.6 | 2.7 | | | | | | | | | | | | | | | | |
| Tam | | 0.1 | 0.03 | | | | | | | | | | | | | | | | |
| u-Yat | | 4.9 | 2.9 | | | | | | | | | | | | | | | | |
| Total | | 102.6 | 61.1 | 39.4% | 43.9% | 46.1% | 24.1 | 26.8 | 28.1 | 24.1 | 26.8 | 28.1 | 100% | 100% | 100% | 39.4% | 43.9% | 46.1% | 43.1% |

Note that the above values are surface catchment based, including both, WAB and EAB. The table indicates the outcrop area of each formation in Wadi Natuf and the respective area rainfall (here taken as area average and seven-year average for the sake of comparison)

