# Peer review of "moisture models in Wadi Natuf karst aquifers, Palestine"

_Hydrology and Earth System Sciences, 2021_

## Referee Comment (RC1)

**General comments**

In this study, the authors apply a methodology based on the landscape classification (geology, soil, land use and land cover) to achieve a regionalization of the recharge. This is done in the Wadi Natuf (occupied Palestinian west bank) watershed which is mostly poorly monitored regarding groundwater (wells) and spring discharge. This appears to be in fact an extension of a recently published work where recharge was estimated at different locations within the basin. The main findings are the identification of land surface features controlling the recharge, here geology, soil thickness, and LU/LC, the mapping of recharge coefficients and an overall recharge rate.

Recharge assessment and regionalization are crucial issues especially in arid to semi-arid domains where groundwater is a major source for water supply. In addition, recharge estimates are rare and difficult to obtain in arid to semi-arid domains which can be seen from the few available values in global synthesis papers. This is exactly the reason why I am generally in favor of studies like the ones presented here, as long as they meet the requirements of a publication in the concerned journal.

I feel uncomfortable with the "novelty" claimed by authors. The use of geological, soil, landscape descriptors for classification purpose to obtain recharge rates (values and distribution) has already been proposed and formalized in published guidelines and/or publications.The authors need to further argue this point and make it clear that such novelty maybe "restricted" to ungauged karstic aquifers.

The manuscript is generally well written, but some parts are unnecessarily long which hinders the fluidity of the reading making it difficult to really capture the ideas and concepts pivotal to the study. This is especially the case for the introduction which looks like more a review. The structuring in subsections in the introduction could be discarded, and the text largely shortened: six sentences l. 29-34 to arrive to the conclusion that the catchment scale is relevant for fluxes identification seems too much for instance. Subsections 1.1&1.3 and 1.2&1.4 could be merged and shortened to give a general context of recharge estimates using landscape characteristics and observations (discharge hydrographs,..).Then the specificity for karstic domains should be more precisely given followed by some information about previous studies and main outcomes in the WAB (section 1.5). Finally, the scope and goals (merging 1.6&1.7) of the present study clarifying the real novelty should be presented. In fact, this is the actual story, but a more concise form would be fine.

There is some confusion or even ambiguity associated with the term "ungauged" as used here. In my understanding, "ungauged basins" refer to watersheds where stream (or spring) discharge is poorly known or even not known at all. Here, the ambiguity is left because in karstic regions springs can be ungauged, the Wadi Natuf can be ungauged as well but the authors mention ""gauged though hundreds of Israeli deep wells" suggesting that gauged/ungauged refers to aquifer monitoring by wells. This needs an early clarification in the manuscript especially if this is claimed as part of the "novelty" of the proposed work.

I found the classification and the transposition into Recharge coefficients very confusing as described in the methodological section. There are some apparent (?) inconsistencies, five classes are proposed but six are used in Figure 5. All this need an extra effort of clarification (see comments 27-28).

**Specific comments**

1) Abstract: The first two or even tree sentences introduce some confusion about the real novelty of the work. Recharges had attracted attention and attempts to provide spatial distributions at local, continental scale or worldwide were made (some reference provided below among others). A distribution over USA is provided in the reference Scanlon et al. (2006) cited by the authors for instance. The same can be said for the reference Zomlot et al. (2018) who provide GW recharge spatial distribution in the Flanders region, Belgium. Unless authors are directly focusing on ungauged AND karstic basins the sentence "relatively little attention has been paid to its spatial distribution" seems somewhat exaggerated. This has to be convincingly stated and argued. In addition, it can be reasonably expected that in arid to semi-arid domains non perennial rivers (wadis) are hardly gauged and the ungauged state is an "internal characteristic".

Baalousha, H.M., Barth, N., Ramasomanana, F.H., Ahzi, S. (2018) Groundwater recharge estimation and its spatial distribution in arid regions using GIS: a case study from Qatar karst aquifer. *Model. Earth Syst. Environ.*4, 1319–1329 (2018).

Jaafarzadeh, M.S., Tahmasebipour, N., Haghizadeh, A., Pourghasemi, H.R., Rouhani, H. (2021) Groundwater recharge potential zonation using an ensemble of machine learning and bivariate statistical models. *Sci Rep*, 11, pp. 5587.

MacDonald, A.M., Lark, R.M., Taylor, R.G., Abiye, T., Fallas, H.C., Favreau, G., Goni, I.B., Kebede, S., Scanlon, B., Sorensen, J.P.R. Tijani, M., Upton, K.A., West, C. (2021) Mapping groundwater recharge in Africa from ground observationsand implications for water security. Environ. Res. Lett., 16, 034012.

Mohan, C., Western, A. W., Wei, Y., and Saft, M.: Predicting groundwater recharge for varying land cover and climate conditions – a global meta-study, Hydrol. Earth Syst. Sci., 22, 2689–2703, https://doi.org/10.5194/hess-22-2689-2018, 2018.

Richts, A., Struckmeier, W.,Zaepke, M. (2011): WHYMAP and the Groundwater Resources of the World 1:25,000,000. In: Jones J. (Eds.): Sustaining Groundwater Resources. International Year of Planet Earth; Springer. doi: 10.1007/978-90-481-3426-7_10

Tillman, F.D, Pool, D.R., Leake,S.A (2015) The Effect of Modeled Recharge Distribution onSimulated Groundwater Availability and Capture, *Groundwater*, 53, pp. 378–388.

2) p. 1, lines 25-26: it can be stated that an average (long-term?) recharge of between 233 and 272 mm yr$^{-1}$ was obtained.

3) p. 1, lines 28-30 : This assertion may be moderated in view of comment 1)

4) p. 2, line 5 "etc" can be removed" or cite the other drivers.

5) p. 6, line 6: in my moderate experience of karstic aquifer, wells are rarely available.

6) p.2, line 36: the term bedrock is usually restricted to crystalline rocks it is useless here. Besides recharge estimates using hydrological water balance (requires surface water information such as stream discharge) or Water Table Fluctuation method ( requires piezometric level information), Scanlon et al. (2006) do not recommend but report methods based on unsaturated zone observations or modeling.

7) p.3, line 7 "lithology" is repeated here, I suppose that "mineralogy" is more appropriate

8) p.3, line 9 Why "land use" is repeated, I was expecting 3 groups with different criteria

9) p.3, lines 6-11: fractures density is not a criterion in karstic domains?

10) p.3 line 20 : "calibrated model parameter": this is too vague what kind of model rainfall-runoff? recharge? other?

11) p 3, line 42 "In ungauged catchment, signatures" : In my understanding, "signatures" such as stream or spring hydrographs are not or poorly available in ungauged catchments (see also general comments)

12) p.4 lines 9-12: Is there a semi-permeable layer in between?

13) p.4 lines 19-20 : if "gauging" refers to wells network, the expression "aquifer wells or piezometer monitoring network" would be more suitable..

14) End of Section 1.5: reader arrives at the end of this section without knowing the main findings of previous studies, no values for the recharge (CMB for instance) are provided..at least orders of magnitude would be useful.

15) p. 4, lines 51-52: please recall the criteria for "a basin classification framework" : land-use, soil, ..?

16) p.5 line 6 :Messerschmid et al., 2020 presumably; repeated many times.

17) p.5 line 15 "different recharge classes were differentiated" according to what king of criterion?

18) p.5, line 34 : "impermeable chalk". This is quite surprising since chalk formations are hardly seen as aquitards. There are no fractures? The use of "low permeability" sounds better for such kind of lithology.

19) p.5, line 36 : 64.4% of the surface area covered by aquifers, what about the remaining fraction Yatta formation? From Figure 2, it is not clear.

20) p.6, line 5 : provide the significance of LBK UBK as given in Table 1.

21) p.6, lines 21-22: Are the springs perennial, seasonal?

22) p.8, lines 44,45: geology, soil and LU/LC were indicated as highly correlated (p. 8, line 17). Is this information not redundant?

23) p. 9, line 10: "chemism" sounds strange, maybe "mineralogical composition"?

24) p. 9, line 12 : "karstic feature" fracturing? Density of fractures…? Please bring more precision.

25) p.10, Table 2 avoid using etc, just mention that it is a generic example and more classes can be used.

26) p. 10, lines 15: It is surprising to read that LU/LC and geology "was not quantifiable in the field". Why? A geological map is available, and authors indicate some modifications from field work. Is there a LU/LC map available? How precisely this missing information was "qualitatively differentiated and correlated", citing the previous study is not enough.

27) p.10, line 21: eight modelled RC values mentioned, six (7 with the zero value, the origin of the value at 49.4% is not given) different values found in Table 4, 5 in bold red in column "2. Soil" of Table 4. This is too confusing, the complete and precise RC values (obtained only for Jer, u-UBK, Heb, l-UBK, and l-Yat after Table 3) should be recalled here.

28) p.10, lines 19-24: It is completely unclear here how precisely "extrapolation" and the transition from Table 3 to Table 4 works: u-UBK receives a recharge class "I" for group 1 and "II" for group 2 in Table 3 but has the same RC in Table 4 for these groups; u-LBK receives the recharge class "II" for groups 1&2 in Table 3 and a different RC value in Table 4.

How precisely these "specific RC-values" were assigned? Is the reader supposed to understand that the different recharge classes (I,..,V) are given the available RC values also known for modelled formations, then the non modelled formations are given the RC corresponding to their class as identified above for each group? If so, it is not clearly explained (see above for apparent inconsistencies).

| Hydro-Fm | Rech. class | RC (group 2 Table 4) |
|---|---|---|
| Jer | I | **57.3%** |
| u-UBK | II | **54.1%** |
| Heb | III | **45.3%** |
| l-UBK | III | **44.7%** |
| l-Yat | IV | **41.8%** |
| Aquitard(Sen,u-Yat, Qat, Tam) | V | **0%** |

Playing with the Table above, I don't arrive to Table 4 from Table 3 in line with the assertion that the transfer is done by group. There is a different but not clearly explained correspondence between recharge classes (I,.., V) and RC values per group. Please clarify how precisely RC *vs* classes relations are built per group: the above table is for group 2, provide the same with explanations for the correspondences in groups 1 (I:54.1; II:45.3; III:44.7; IV: 41.8; V:0) and 3 (I:57.3; II:54.1; III:**49.4?**; IV: 41.8; V:0).

The above table is slightly inconsistent (for class III) with the color bar legend in Figure 5 which unexpectedly accounts for "a 6[th] extra class".

29) p.10, line 24, insert "(see Table 3)" after "formations".

30) Section Results: Except the global recharge, this section brings little additional information in comparison to the methodological section. A regionalization is presented Figure 5 for group 2. What is done with the other groups? The resulting map for the three groups could be represented in a 3 panels figure with the corresponding color scales.

31) p12, line 6: "aquiferal" first time I read such formulation, "aquifer" is enough.

32) p. 13 lines 9-22: Is this supposed to be a discussion or an introductive state of art?

33) p.13, line 21: what is a "moderate climate" temperate?

34) p.13, line 24 : " This study went a step further" of what? It is unclear.

35) p 13 line 33 to p. 14 line 5: This is a too lengthy preamble for an equifinality problem that is not really addressed here or a correlation issue between descriptors.

36) p.14, line 5-8: Is really equifinality addressed here or the uncertainty on recharge distribution. Equifinality problem would suppose that several recharge maps could reproduce some observation of the average recharge for instance (which is likely!).

37) p.14 lines 26-28: I don't find any elements supporting this discussion in the manuscript.

38) End of section 5.1: arriving at this stage, no word about the limitations or caveats of the approach which can be expected in a discussion.

39) Section 5.2: Maybe the following can be of some use..

Average recharge coefficients of between 23 and 52% (average 35%) for karstified carbonates (Allocca et al., 2014; Arfib and Charlier, 2016; Baudement et al., 2017; Martos-Rosillo et al., 2015; Polemio, 2016; Zagana et al, 2007), and between 10 and 21% (average 16%) for detritic aquifer formations (Seraphin et al., 2016; Yagbasan, 2016; Zagana et al, 2007) were reported in the north Mediterranean area (Spain to Turkey).

Arfib B., Charlier J.B., 2016. Insights into saline intrusion and freshwater resources in coastal karstic aquifer using a lumped Rainfall-Discharge-Salinity model (the Port-Miou brackish spring, SE France). J. Hydrol. 540, 148-161.doi:10.1016/j.jhydrol.2016.06.010.

Baudement C, B. Arfib, N. Mazzilli, J. Jouves, T. Lamarque, Y. Guglielmi, 2017. Groundwater management of a highly dynamic karst by assessing baseflow and quickflow with a rainfall-discharge model (Dardennes springs, SE France), BSGF – Earth Sciences Bulletin 188, 40 https://doi.org/10.1051/bsgf/2017203.

Martos-Rosillo S., Gonzalez-Ramon A., Jimenez-Gavilan P., Andreo B., Duran J.J., Mancera E., 2015. Review on groundwater recharge in carbonate aquifers from SW Mediterranean (Betic Cordillera, S Spain), Environ Earth Sci 74, 7571–7581. https://doi.org/10.1007/s12665-015-4673-3

Polemio M., 2016. Monitoring and Management of Karstic Coastal Groundwater in a Changing Environment (Southern Italy): A Review of a Regional Experience. Water 8, 148, doi:10.3390/w8040148.

Séraphin P., Vallet-Coulomb C., Gonçalvès J., 2016. Partitioning groundwater recharge between rainfall infiltration and irrigation return flow using stable isotopes: The Crau aquifer. J. Hydrol. 542, 241–253.

Yagbasan, O., 2016. Impacts of climate change on groundwater recharge in Küçük Menderes River Basin in Western Turkey, Geodin. Acta 28, 209-222, Doi:10.1080/09853111.2015.1121802.

Zagana, E., Kuells, Ch., Udluft, P., Constantinou C., 2007. Methods of groundwater recharge estimation in eastern Mediterranean - A water balance model application in Greece, Cyprus and Jordan. Hydrol. Process. 21, 2405-2414, doi:10.1002/hyp.6392.

40)p.14, line 16 "geology" as it appears in the manuscript and not "lithology"

---

## Author Comment (AC1)

Dear Reviewer. Thank you very much for your insightful and detailed comments and recommendations. Please allow us to answer as follows:

**ANSWERS**

**Specific comments**

a) Methodology. Wadi Natuf, prior to our work, was not "mostly poorly" but entirely ungauged. No rainfall, runoff, spring flow records, let alone soil moisture data were available.

b) We concur, thank you.

c) Novelty of our approach. Indeed, many, if not all of our individual tools and methods applied for Wadi Natuf are standard, well tested, used and published. Two things make our approach novel: First, a fully distributed approach was followed. Second, this is one of the many ungauged basins worldwide, which however are underreported in the literature. We specifically follow the recommendations of the IAH decade on PUB, which have described and prescribed in general terms but not been implemented before under such circumstances as found in wadi Natuf. (Most PUB applications deal with runoff, rather than recharge. Exceptions that tackle recharge, such as *Aish et al.* in Gaza, deal with gauged basins, where large amounts of subsurface and water level data are available and drawn upon in the study.) Our paper is a strong departure from this state of the art. And thirdly, maybe most importantly, it is the unique combination of individual tools, their application and synthesis in this one basin framework of ours, which is novel. Fourth, our results base on empirical field work, not calibrated models. So, yes, we agree with your comment that one novelty is the work in ungauged basins, but the other two factors should not be forgotten or underestimated either. We will try to make this more clear (abstract, introduction and conclusions). [If you are interested in a detailed discussion of each of these articles, on the *already proposed and formalized values of distributed recharge*, we can send you an additional file upon request.] At the end of our rewritten chapter Introduction we list seven different points that – taken together – characterise the novelty of our approach: The model is fully distributed, not lumped or semi-distributed [1]. It is a recharge model, not simply a rainfall-runoff model as in other studies before [2]. It concerns a hitherto ungauged basin [3] of deep karstic aquifers [4]. *It employs a new combination of* existing techniques that are based on observable processes, parameters and signatures with the help of a specific basin classification framework [5]. Our assessment adheres to the goal of parsimony [6] and integrates inductive and deductive steps. And, with the observation of processes deep underground being made impossible (by the occupation), it is grounded in surface and surface-near observations (so-called direct approaches) [7].

d) Length of the manuscript. Thank you for this comment and your recommendations. Partly the length and detail of the introduction is the result of the earlier reviews of the first manuscript, which then was split up into two different articles (This draft, reviewed by you, forms the second of the two articles.) We will try to shorten, condense and reduce the amount of information in the Introduction. (Real cuts cannot be done without also dismissing some of the content of this chapter.) So, a compromise will be necessary and attempted by us.

e) Ambiguity of the term "ungauged" – Dear reviewer, indeed, we also define and understand "ungauged" as lacking any flow or other hydrological information (incl. meteorology). We do not speak about special cases in karst, where some gauging data may be available. Where refer to "hundreds of wells" – deep Israeli wells – that are gauging the basin, we specifically refer to the downstream Coastal Plain area of the basin inside Israel (in the borders of 1949) only. By contrast, the upstream recharge areas, located in the occupied West Bank remain fully undeveloped, untapped by wells and thus ungauged. This is the study area of Wadi Natuf. All of Wadi Natuf was ungauged, prior to our study and field work. Such contrasts between gauged and ungauged regions with a basin are not only very common worldwide, but are especially relevant for the question of distributed recharge, which has to be investigated in the – typically ungauged – portions of otherwise already developed and used aquifer basins. In the case of Israel and the West Bank under prolonged military occupation, this contrast is particularly stark and it was specifically the underlying motivation of this study to try help close the gap, if not development and use, then at least in investigation, research and understanding.

f) "Classification and transposition into Recharge coefficients". Thank you, yes, we understand your point and why there seems to be some inconsistency in the way we portrayed our methods and approach. We will clarify this, including changes in the tables presented… (see also, Specific Comments).

**Specific comments**

1) "*Abstract: Unless authors are directly focusing on ungauged AND karstic basins the sentence "relatively little attention has been paid to its spatial distribution" seems somewhat exaggerated.*" → As already discussed in the general comments, the novelty of our approach is the novel COMBINATION of otherwise well-established methods and procedures. We combine empirical field work with deductive methods. We work on an entirely ungauged basin – ungauged in any form, whether with regard to precipitation, runoff, spring flow, soil moisture or recharge. (The only exception is one single deep groundwater well under Israeli control, well Shibteen No. 4). Hence all our measurements were a first step into the unknown. But importantly and in addition, the novel character of our work stems from the fact that we attempted an assessment and quantification of GWR in its spatial distribution: In contrast to the literature suggested by you, our study does not only weigh and qualitatively evaluate certain "descriptors" or "physical factors" of recharge or estimate recharge distribution in relative terms or as "zones" of (weak, moderate or strong) GWR, but actually presents a fully distributed quantitative recharge model and assessment over the entire catchment area.

As to the quoted phrase "relatively little attention" – we would like to stress that the emphasis in this sentence lies on RELATIVE. Indeed, spatial distribution of GWR and its quantification has been much, much less studied than its temporal variability. This is something we fell, we cannot change (only clarify to avoid misunderstandings).

Concerning karst, and although most aquifers are indeed karstified, our method of 'direct procedures' tries to understand and quantify the processes of recharge, where they are accessible at the surface (without boreholes or other groundwater information in the underground). As a consequence, our SM-measurements and modelling takes place above the aquifer, at the soil layer and its interface with the underlying aquifer bedrock. Hence, the typical problems of describing the characteristics of karst (such as anisotropic flow) could be circumvented. Therefore, our model is applicable also in karstic rock, but as well generally in all kinds of aquifer types.

First, we will change the manuscript and state more clearly that the novelty of our approach lies in the combination of techniques, used in this study. Secondly, we will somewhat caution and specify our statement, that so far relatively little attention has been paid to spatial GWR assessment and quantification (and in a fully distributed model).

Last, regarding the literature suggested: Thank you very much! Some of the texts were indeed new to us (or were published after our work). However, they do not change our views expressed in the paper. We would argue that each of them falls short of such an approach as followed in our study (or does not even attempt to answer the same questions). We attach a paper with some discussion of the individual studies.

2) P. 1/26: Yes, thank you! We will indeed add a more comparable quantification of our results in mm/a (235 -274 mm/a).

3) P. 1/28: Following the above (1), we would still maintain that this COMBINATION of methods was used for the first time in such an environment and for such purposes (quantifying distributed GWR…). We would also like to stress that this second part of our series of articles on Wadi Natuf recharge cannot be read and understood without the first article. Much of the empirical field work is described in article 1. This second part on the regionalisation of the results of the study (in article 1) employs more deductive methods and conceptual work on the BCF. We were advised to split the originally submitted overall manuscript into two articles (both by reviewers and editors of HESS) – this 2nd part (here under review) is the result of that split.

4) P.2/5: Yes, thank you. We will remove "etc." and change the manuscript into: "their drivers, e.g. precipitation **and** evapotranspiration (Batelaan… "

5) P.2/6: We beg to differ: Throughout the Levant, the mostly Upper Cretaceous/Lower tertiary aquifers are all karstified. And they are all heavily developed, tapped, drilled and abstracted in all states – Lebanon, Syria, Israel and HK Jordan (ESCWA-BGR; 2013: 285) – with the notable exception of Palestine (the occupied West Bank of Wadi Natuf), where well development is rigorously restricted by the Israeli hydro-regime of its military occupation… As stated further below, in the same aquifer, the WAB, the Israeli downstream side is heavily developed and used, while the upstream occupied West bank zone (= Wadi Natuf) remains in an enforced virgin state of well development. So, in all karstic aquifers of the region, karstic aquifers wells are available in abundance – except for the West Bank. (Sentence remains unchanged.)

6) P. 2/36: We have found the term bedrock in the literature not only describing crystalline bedrock formations, but also as a more general term (Nature, USGS, etc.). However, we can change this term if required.

Second, yes, thank you. We change the sentence. Scanlon et al. report rather than recommend such methods (unsat. zone observations).

7) P. 3/7: Geology in our view encompasses more than only lithology. Also, the quoted authors (Sanz et al., 2011) differentiate somewhat between the two. The second mentioning of "lithology" refers to another set of studies (Batelaan, Aish, etc.). It is not our repetition but theirs. (Please note, as already mentioned, we here not only list the three groups but also forms of combination and interaction with each other. Repetitions therefore are unavoidable.)

8) P. 3/9: Again, land use is not simply repeated as a mistake. Rather, the different authors use different approaches, i.e. different combinations of factors (like land use together with other

factors). We wrote: Authors use three principle groups (and use one or two of them). We can clarify the distinction between 3 principle groups and their combination by different authors.

9) P. 3/6: Yes, but do we understand you correctly that you want "fractured density" to be mentioned (as was used by Radulovic)? However, our aim here was not to be as exhaustive as possible in a list of criteria. Rather to the contrary, we aim for simplification and here want to show the three principle groups that are used most frequently. → Is it necessary to add a sentence that lists all other possible criteria? For example, to determine "fracture density", you usually need very extensive ground work, mapping and even underground information. This can never be done for an entire area or catchment, but only and at the best, at some isolated spots in the field... – and such preconditions are very far from Predictions in Ungauged Basins (PUB)!

10) P.3/20: Yes, thank you. We could have specified the type of model: "*In his **rainfall-runoff** model, Seibert (1999) developed relationships between the calibrated model parameters and the physical catchment characteristics of landscape found in the field*". However, since you asked us to shorten the Introduction, we removed this sentence from the manuscript.

11) P.3/42: Thank you. You are right, where basins remain completely ungauged, it is often difficult to find and identify any such signatures. So, signatures require some, if even poor gauging beforehand (or of course field observations and measurement campaigns during a study). Yet, despite this fact, this is what the quoted PUB-authors recommended. We could alter the sentence slightly and add: „*Although, such signatures are often lacking due to the state of remaining ungauged…*". Would that solve the problem? In our case of Wadi Natuf, after and due to the measurement campaign of our study, we were able to quantify such signatures. The patterns of such signatures were already detected before measurements began, during the stage of field observations and therefore, measurements were designed and carried out accordingly (see also part 1 of our article series). Your comment was considered and has changed in our new, shortened version of the Introduction.

12) P. 4/9: Thank you, we will amend this sentence into: The WAB is **a complex of** up to 1000 m thick Upper Cretaceous carbonatic karst aquifer (SUSMAQ, 2002) and conventionally divided into two regional aquifer layers (Fig. 1) – an Upper Aquifer (UA) of Turonian to Cenomanian age and a Lower Aquifer (LA) of Upper Albian age, (see Fig. 2a in Messerschmid *et al.*, 2019) **– and separated by poorly or non-permeable layers of Lower Cretaceous age (Yatta formation)**.

13) P. 4/19: Ok, we will change into: …monitored and gauged through **a network of hundreds of Israeli groundwater abstraction and monitoring wells (tapping the deep aquifers). By contrast, the slopes** , the WAB recharge and accumulation zones **(inside the occupied West Bank)** remain …

See also general comments

About your comment: "*In my understanding, "ungauged basins" refer to watersheds where stream (or spring) **discharge is poorly known** or even not known at all. Here, the ambiguity is left because **in karstic** regions springs can be ungauged, the Wadi Natuf can be ungauged as well but the authors mention ""gauged though hundreds of Israeli deep wells" suggesting that gauged/ungauged refers to aquifer monitoring by wells.*"

We shall explain here that the WAB is well gauged in its **discharge** area, but that almost the entire **recharge** area (and Wadi Natuf) remains completely ungauged (groundwater flow and discharge of springs is not known at all) = PUB.

14)    P. 4/End of section 1.5: Thank you for this comment. Indeed, we did not intend to already introduce results of other studies, not least in order to avoid unnecessary repetition (which you criticised on other occasions) and since later, under results and discussion we will come back to this point. It should also be mentioned that in part 1 of the series, we presented findings from other studies extensively (see Table H1 in the Appendix). Maybe, it would suffice here to make reference to this article and Table of comparison (see: https://hess.copernicus.org/articles/24/887/2020/)? As for the results of THIS study ("values for recharge"), we do not think it is proper and necessary to already pre-empt the final results of our study in the middle of the chapter Introduction. So, we only briefly added here a RC range of 30-50% (together with a reference to Annex H of the first paper, Messerschmid *et al.*, 2020).

15)    P. 4/51: We would like to state that we cannot mention our BCF criteria in this sentence about recommendations by Hrachowitz. Hrachowitz, in his study did not specifically mention these criteria. To the contrary, the combination in our BCF is our novel combination of such criteria, as will be presented at the appropriate time (methods and results of the study), not already here. Therefore, in order to follow your recommendation, we can only mention again the three groups (but not as groups already identified by Hrachowitz).

16)    P. 5/6: Yes, you are right! This is now Messerschmid et al. 2020! Thank you.

17) P. 5/15: Question: Differentiation of recharge classes. – based on which criteria? Answer: The classes were differentiated in two steps: A) Qualitatively, based on observable features and well-established physical principles of recharge processes (for example difference between karstic carbonates and clay; or between forests and sparsely vegetated land forms, and of course between thin and thick soils).  B) Quantitative differentiation was based directly on the results of the SM-percolation models (in Messerschmid et al., 2020). See new version of the Introduction.

18)    P. 5/34: Impermeable chalk. Thank you – we understand your concern. Indeed, in other areas of the world, "Chalk" is a term for extremely powerful, productive aquifer formations, such as famously in the case of the "British Chalk" – a microporous white limestone formation, which however exhibits rather highly competent rock, in which fractures and fissures remain open and therefore act as excellent flow paths! (see Introduction by BGS - The Chalk). However, in the Levant, the Senonian chalks are almost invariably known as powerful aquitards (even dubbed "aquicludes" here). This is because this chalk is very soft and incompetent. Hence, it does indeed act as a powerful separation layer between over- und underlying aquifers (Eocene and Turonian aquifers), both on a regional and local scale. Few exceptions of some minor aquitardal limestone series within the Senonian formations actually prove this rule. Therefore, in general, as well specifically for Wadi Natuf, we would find it rather misleading to attest a "low permeability" to the Senonian chalks, rather than stressing the completely "impermeable" and separating nature of these aquitards ("aquicludes")…

- "The thick **chalk-limestone** Senonian to Eocene sequence is generally an **effective aquiclude**" writes **Carmi** (1989): https://www.sciencedirect.com/science/article/abs/pii/0022169489901698
- see also:  **Avisar** et al (2004), Fig. 2 (modified after Rosenthal et al. 1999) - https://link.springer.com/article/10.1007%2Fs10040-004-0322-8
- see also works of Akiva **Flexer** (TAU): https://link.springer.com/article/10.1007/s100640050030
- and **Qannam** (2003) in Al-Arroub: https://tu-freiberg.de/sites/default/files/media/institut-fuer-geologie-718/pdf/fog_vol_9.pdf
- and **Sauter** et al. (2005), Final Report of GIJP: "*The formations of the Mount Scopus group can generally be considered as an aquiclude.*" https://www.cleaner-production.de/fileadmin/assets/bilder/BMBF-Projekte/02WT0161_-_Abschlussbericht.pdf

19)    P. 5/36:   One has to read this statement carefully and correctly (see below). First, we refer to Fig.2b in the previous study, not in this manuscript under review (we do not have a Figure 2b in this manuscript). By contrast, Fig. 2b in the previous article clearly indicates the entire list of existing lithostratigraphy (top left corner with legend in the map). Figure 2 of this manuscript has nothing to with it, as it is meant to present the correlation between LU/LC and geology… Second, we are speaking of the WAB portion of Wadi Natuf only. Thirdly, we present the "conventional view", i.e. only two large regional aquifer complexes, UA + LA. (UA = *Heb, l-Bet, u-Bet & Jerus.*; LA = both Beit Kahil formations, or *l-LBK, u-LBK, l-UBK & u-UBK*). This is NOT the more refined hydrostratigraphy as applied in this study. It is only a brief summary introductive statement about the geology and hydrogeology of the WAB in Wadi Natuf. Roughly two thirds of outcrops of this WAB portion are made up by the conventional LA and UA formations (64,4%). In other words: This does explicitly NOT include the aquiferal portions of middle Yatta formation (as it is considered *The Middle Aquiclude* in the conventional view). Also excluded here is the deep Ein Qinya formation aquifer outcrop portion as well as the shallow alluvial deposits and their share of total WAB are in Wadi Natuf (not part of the "Mt. aquifer" stratigraphic column). For more details, see further explanations in part 1 of the article series. The exact outcrop sizes of each formation are detailed in addition further down in Table 4 (km$^2$ outcrop of each formation) for the entire Natuf catchment (incl. the Eastern EAB portion). Now, the text is slightly adjusted into:

All formations of the WAB are covered in this study (Fig. 2b in Messerschmid *et al.*, 2020). Together, the aquifer formations cover around two thirds (64.4 %) of the outcrop areas in Wadi Natuf; they are entirely carbonatic and in most parts strongly karstified. The other third of the area consists of outcrops of less permeable and fully impermeable formations.

20)    P. 6/5: LBK & UBK comment: "*provide the **significance** of LBK UBK as given in Table 1.*" It is not entirely clear what you mean by that. Are you asking, a), why LBK and UBK form a *major "regional aquifer*" in Tab.1? Or do you ask, b), why they are differentiated as in Tab 1? Or c), is it the word "must"?

*Answer to a)*, LBK (+ UBK) are well known and used as main target aquifer in all deep wells (most productive ftn.) in the West Bank.

*Answer to b)*, l-UBK differs strongly from u-UBK or l-LBK (this is also a practical problem, felt in many deep wells, where these differences were described in terms of different lithology, fracturedness, etc.) And to c), of course, this can be changed: On a local scale, the aquifers CAN be differentiated into different (more or less permeable) parts.

21)    P. 6/21: Yes, most of the springs are perennial. And a few (some 10-15%) of them are very small springs that dry out seasonally each summer). More details on springs are found in part 1 of the article series.

22)    P. 8/44: Yes, indeed, this correlation is repeated several times throughout the article (like before on page 8, line 17,18). We thought this is a central fact worth emphasizing. But we can also avoid such repetition…

23)    P. 9/10: yes, fine. We will change into "mineralogical composition".

24)    P. 9/12: Fine, no problem. If that helps, we can list these karstic features studied in the field: The text can be amended as follows: "First, existing geological maps in the scale 1:50,000 (GSI, 2000; 2008; Rofe & Raffety, 1963) were corrected, complemented and refined by extensive field mapping and

remote sensing (stereoscopic aerial photographs) with the target to detect, describe and interpret the lithological rock content, (chemism, texture, grain distribution), the degree of crystallisation and structural features like folding, faulting, cleavage, jointing, as well as primary porosity and karstic features. These karstic features encompass **epi-karst surface features** (karren and schratten landscape), **karstic solution holes** (especially in the l-LBK formation, aka "Swiss cheese" formation and in parts of the Hebron formation), well-known **caves and karstic channels** in the underground; wide, **karstified fractures** (incl. their width and prevalence). In addition, there are **indirect indicators** of sub-surface karst (such as rapid interflow emerging at "bleeding hills" and travertine crusts)."

25)   P. 10/2: Ok we will remove the line with "etc." from Table 2 and will amend the text in line 18 as follows: "were inserted 17 in Table 2 **as a generic example, in order** to obtain…"

26)   P.  10/15: Thank you for this comment. We can see now that our statement was confusing. What we wanted to say is that both LU/LC and geology are not quantifiable parameters in the sense that forest and grassland or limestone and chalk are not quantitative categories. But of course, the area size of their occurrence can be and was quantified. And in addition, quantitative values can be attributed to these qualitatively different features if needed. In fact, we did attribute different RC-values to these land use features and geological formations as a result of our BCF. We should amend our text here and rather state that **LU/LC & geology features were ranked according to their recharge potential and correlated with soil thickness** (see: ranked classes of recharge potential from low to high, as in Table 2). **This was based on a soil thickness correlation matrix in which representative typical soil depths could be attributed to the different formations and land forms** (this soil matrix is shown in Appendix D, Messerschmid et al., 2020).                  Regarding your second question, yes, a LU/LC map is available and presented in Figure 2 (p. 7).

27)   P. 10/21: Thank you. Yes, indeed - maybe we should formulate it more simply and straight forward without unnecessary details: "Five different formation-specific RC-values were obtained from the monitored and modelled SM-station data (Messerschmid et al., 2020), representing 5 different classes of recharge potential (between 57% and 42%). In addition, impermeable formations were not monitored at SM stations. For them a sixth class of zero recharge (RC = 0%) is added. According to the correlation and grouping in the BCF, the different specific RC-values of the modelled formations could be attributed to other formations as well (see Table 4). By this step, all existing formations in Wadi Natuf were assigned specific RC-values of annual recharge (ranging from 57% to 42% in aquifers and down to zero recharge in the aquitards). It should be added here that for three formations, an additional step was needed. These formations were found not to be uniform but either consist of different sub-facies at different locations of the catchment (e.g. u-Bet and l-Bet) or represent different lithologies in the stratigraphic column (within l-UBK formation, see photo, Fig. 3c, above). In these instances, an additional, intermediate RTC value was introduced, based on the arithmetic mean of classes II and IV (49.4% - as average between 44.7% and 54.1%)."

28)   P. 10/19: transition from Table 3 to Table 4: Thank you for your comments. You are right, the text and the tables are somewhat confusing. We added information and restructured Table 3. An additional "class" of recharge coefficients was inserted, not based on direct results of SM-percolation modelling, but as an average of two of the classes calculated by this modelling. (classes II and IV → new average: 49.4%). This applies only to the group (column) "geology" and it is based on the fact that the three formations concerned all display a mix of different facies (either vertically in the rock column or laterally in the outcrops of the catchment). We therefore have to also adapt the text of the manuscript accordingly (see answer to comment 27), above).

Table 3. Conceptual basin classification framework, specific for Wadi Natuf

| | group 1 - LU/LC | | group 2 - soil | | group 3 - geology | | |
|---|---|---|---|---|---|---|---|
| | formations | features | formations | feat. | formations | features | |
| I | | | All, **Jer**, l-LBK | -- | All, Jer, l-LBK | well dev. karst (& gravel) | |
| II | u-UBK | cliff, mostly rock outcrops | **u-UBK**, u-LBK | - | Heb, u-UBK, u-LBK | karstified lst / dol | |
| * | | | | (…) | l-Bet, u-Bet, l-UBK | lst / dol (some marl / chalk) (*Nari for u-Bet*) | Increasing recharge potential ↑ |
| III | All, Jer, Heb, u-LBK, l-LBK | olive terraces, rock outcrops | u-Bet, **Heb**, EQ | -/+ | | | |
| IV | u-Bet, l-UBK | arable but uncultivated, grass- & shrublands | **l-UBK** | +/- | | | |
| V | l-Bet, l-Yat, EQ | mixed, transit. woodlands | **l-Yat**, l-Bet | + | l-Yat, EQ | mixed lst + marl | |
| - | *(as Gr.2)* | agric. plains, forests | Sen, u-Yat, Apt | ++ | *(as Gr.2)* | marl (chalk) | |

*Note*: Left column: classes of measured recharge potential (I – V); middle columns: groups of phys. features (1-3); formation names as in Table 1; soil thickness increases from thin (--) to thick (++). The formations shown in bold type were the ones monitored, measured and modelled. The grouping and class distribution was based on field work and literature, e.g. SUSMAQ (2002), LRC (2004), GSI (2001), Keshet and Minran (1993), Messerschmid (2014) and Messerschmid *et al.* (2018). Aquitards, i.e. impermeable formations, where recharge is assumed zero, were not measured in SM-stations (bottom line of Tab. 3).
*Regarding class* **\***: The class marked with asterisk **\*** was not measured in SM-stations. Instead, for group 3 (Geology), the average of RC for classes II and IV was taken (as 49.4%), because these formations appear in two facies types, which are more and less permeable, respectively.

29) P. 10/24: Ok, thank you – we will insert reference to Table 3.

30) SECTION RESULTS: We are not sure what precisely you suggest. Do you want us to transfer some of the information from Section 3. Methods to section 4. Results? (*NB: We re-formatted the numbering in RESULTS to 4.1, 4.2, etc.*) otherwise, fine, we amended Figure 5 and added a map for LU/LC and for Geology; it now shows all three different runs for all three groups.

31) P. 12/6: Fine, we can change into "aquifer" here.

32) P. 13/9: Discussion often starts be recounting the state of art (referring back to the introduction and thus introducing the relevant questions for discussion). Admittedly, this is a matter of personal style and taste. But many articles follow and many tutorials on scientific writing suggest such a procedure. We can of course also simply cut the entire paragraph (line 9-22), if this is really what is demanded by you and the other reviewers and editors.

33) P. 13/21: Oh, yes, of course – thank you! "temperate" not "moderate"…

34) P. 13/24: Yes, thank you. We will reformulate, but what we intended to say here, and referring to the recount of existing works in lines 9-22 before, we wanted to state that our study went a step further *than* **the existing literature**, a step beyond the methods employed already in the quoted literature. This is part of the novelty of our approach. Again, the novelty here is– to the best of our knowledge – the unique COMBINATION of otherwise well-established methods. But this, by the way, is not the only novelty. Other new features are the application in hitherto completely ungauged basin, the strictly empirical nature of our approach (and its depth), as well as the truly distributed recharge analysis, which hitherto has not been performed under such circumstances (karstic, PUB, recharge, not runoff calculations, forward calculating models, not calibrated and retro-fitted modelling, etc.)

35) P. 13/33 ff.: Thank you. We will shorten this paragraph, as follows: "Our three independent runs of conceptual analysis and attribution resulted in a close range of total WAB recharge (24, 26 and 28 mcm/a). This suggests that our transfer procedures delivered a robust and realistic representation of the processes at hand (which we prioritized over allegedly "exact" but less reliable results)."

36) P. 14/5: And hereby, we will focus on the point that and why three different runs were carried out separately for each group of physical features.

37) P. 14/26: Yes, you are right. It should read: "*The results **of the measurements and analysis in part 1 of the series confirm** the well-known fact that the temporal distribution of events (precipitation) strongly affects the percolation rates.*" We also wanted to remark on the side that our daily steps of analysis were the appropriate time scale; we thought this a relevant

issue under the topic of "process representation" (section title 5.1) – but if you deem this unnecessary in this paper, we can also simply delete this statement…

38)    End of Section 5.1: Yes, indeed, limitations and caveats are discussed at the end of the Section Discussion, not at its introduction. → We added a new section No. **5.2 Limitations & caveats** (followed now by a new section No. "**5.3**" for "Annual RC – overall basin RC …")

- To begin with, the results of the SM-models, the RC-values of the different formations as input data for our basin classification framework (BCR), are taken as correct and reliable. A discussion of the limitations and caveats of the results and methods can be found under Messerschmid et al. (2020).
- However, the process of setting up a BCF and attributing different classes of recharge potential (RP) to the different physical features (under the 3 groups selected) is a deductive step, which relies on the translation of qualitative observations in the field into quantitative classes of RP. Therefore, the exact classes under the here developed BCF, although based on and rooted in universal physical laws and well-established evaluations, could be somewhat imprecise and incorrect.  Some classes could have been selected wrongly and could under- or overestimate certain factors (features) for the decision. This is why we found it imperative to establish three independent runs of classification for the three different groups of indicators, which allows us to weigh and compare and thus evaluate the reliability of the BCF.
- Another theoretical possibility is that some processes, although present in the field, were not detected and included in the set-up of the BCF. However, the approach of this study used the most commonly known principle groups of physical landscape criteria quoted in most of the literature (see Ch. 1, Introduction). Therefore it can be stated with confidence, that the processes covered by our selection of classification criteria belong to the most important, principle processes of GWR and it is rather unlikely that a major process was overlooked.
- The possibility of overlooking a minor process is always and necessarily a by-product of such simplification. Hence, such simplification, a major characteristic of our approach is not only a strength but also a (relative) weakness. However, it should be repeated here once again that the need for simplification of the host of processes at work in groundwater recharge is strongly recommended and explicitly highlighted by the existing PUB-literature (see Ch.1, Introduction).
- In addition, the overall results were also weighed against and compared with similar results from other catchments, especially such in in the WAB and its environs.
- Lastly and although the correlation between the three groups of observable features was clearly observed and investigated in depth within Wadi Natuf, it may be absent in other catchments. This then would pose a limitation to the applicability of the approach chosen. However, in such a case, other correlations can and should exist; they should be studied and detected individually for each other basin, but otherwise following the same approach as designed for this study.

39)    Section 5.2: "Literature on RC values in other areas…": Thank you. The studies of Allocca und Marto Rosillo are useful. They **shall be briefly mentioned and their avg. RC values quoted**. But in the other studies we cannot see the applicability for our purposes [further details and discussion are available upon request]. Hence, we **added** to the text and references:

**Allocca** et al. (2014) quoted average annual RC-values ("effective infiltration") from other countries (Hungary, Greece, Spain, France and Croatia) to range from **35% to 76 %** and of 27 % for Tennessee (dolomites) and **found** recharge coefficients of **50 – 79 %** in their own study in the southern Apennines. **Martos-Rosillo** et al. (2015) present a review of gw recharge studies in Spain. They found spatial variations due to: "*the degree of surface karstification and the development of the vegetal cover–soil–epikarst system in the carbonate aquifers*". "*The recharge may range anywhere from 7 to 720 mm/year. The mean coefficient infiltration or recharge rate is 38 % of the rainfall, ranging between 4 and 62 %.*"
[**Added to References:** Martos-Rosillo, S., González-Ramón, A., Jiménez-Gavilán, P., Andreo, B., Durán, J. J., & Mancera, E.: Review on groundwater recharge in carbonate aquifers from SW Mediterranean (Betic Cordillera, S Spain). Environ Earth Sci, 74(12), 7571-7581, 2015. https://doi.org/10.1007/s12665-015-4673-3 ]

40)    P. 14/16: yes, thank you. We will write "geology", not "lithology"…

Clemens Messerschmid, Amjad Aliewi (Dec 2021)

---

## Author Comment (AC2)

Thank you very much for your review, insightful comments and helpful recommendations.
Please allow us to address your general and specific comments in the following.

**General Comments**

1. **Introduction (too long)**

This comment was also made by Reviewer No. 1 and we changed, reorganised and shortened our Ch.1 Introduction accordingly.

2.

3. **Quantitative aspect of RC attribution**

Also this was already found in review 1 as a main shortcoming of our manuscript. We adapted our manuscript thoroughly, changed Table 3 and added more precise descriptions and explanations of our work.

4.

5. **Societal issues**

We slightly adapted and added to our manuscript (Section 1.3)

6.

7. **Climate change & emphasis on applicability to many major resources of the Mediterranean region**

We added a short mentioning of climate change under Ch. 6 (end of first sentence; third sentence).

**Specific Comments**

8. **Again: Length of Ch. 1 Introduction**

As in 1. Above.

9. **Discussion of application to other peri-Mediterranean sites (soil, crops, geology…)**

8 lines of text were added to Ch. 6 Conclusions (sentence 3 ff.)

10. **Again (and as in Review 1): Quantification of RC-values (% figures)**

As noted before, major changes were made in the manuscript. The methodology was explained more precisely and the text and Table 3 in the Ch. 4 Results were changed.

And more specifically on your remark: "This point [*percentage value*], which makes it possible to quantify the recharge areas and to propose a detailed map, is not explicit at all." Major changes were made in Ch. 3 Methodology.- The quantification is now better explained and stated more explicitly in several of its sections (for example 3.2.). We explain, where the quantitative RC figures come from (soil models in the first article of the series) and how they were used for other formations under the Soil Group, as well as attributed to the other groups (and their different formations). Please note, as already discussed in our answer to Review 1: In all but one formation, we used the exact values of RC as modelled at the representative soil measurement sites (for 5 different formations) – see first article (Messerschmid *et al.*, 2020). The one exception (l-UBK formation), where we introduced a new RC (composite value of two modelled RC-values), is now highlighted an explained in both text and table 3.

11. **The role of slope in recharge and its role in our Basin Classification Framework**

This is a valuable comment, thank you. In principle you are right and indeed, also relief (slopes) plays an important role. But in our specific case we opted for not singling it out as a distinct Group of our

Basin Classification Framework (BCF). It was however included as one of the relevant factors in our analysis on land forms (LU/LC Group).

More specifically: The matter of slopes - especially terraced slopes and their role in soil infiltration and percolation - had been dealt with in the first article. For this purpose, their nature and distribution were observed in the field, and their typical appearance and prevalence was attributed accordingly to the different lithostratigraphic formations that crop out in Wadi Natuf (together with the typical soil thickness cover over each such formation). The results were discussed and presented in the first article; see our "matrix" in Table D1, Annex D in Messerschmid *et al.* (2020). In this new manuscript (under review), this matrix is mentioned 4 times in the Methodology and another time in the Discussion. In the new article under review, this correlation is also mentioned in the text and shown in the photographs in Fig. 3.

Please allow us to go into more detail on this point (see discussion at the end of this file).

**Technical corrections**

**12. Colours in Figure 1 and 2**
The colours were adapted and unified

**13. Colours in Table 1**
Please note that the colours in Table 1, in our opinion, should not be the same as in Figures 1 and 2. The figures are area-specific maps of the different formation outcrops in the area - grouped as main regional aquifers and aquitards: Upper Aquitard (Senonian) / Upper Aquifer / Middle Aquitard (Yatta) / Lower Aquifer / Bottom Aquiclude (Aptian). By contrast, Table 1 not only lists the Regional hydrostratigraphy but also the much more refined local hydro-stratigraphic divisions, which do not appear in the Figures. We would find it difficult to apply this regional colour code to the local fine-stratigraphy. In addition, the main purpose of this column is to portray the general conceptual differentiation between formations of different aquifer potential (major / minor / good / poor /local and none), which applies to all aquifers/aquitards worldwide. Our purpose here is that and to which extent the regional stratigraphy subdivides into a more refined pattern on the local scale. Using a rather chronostratigraphic colour code for the different formations (and sub-formations) would rather create some distraction from the main purpose, at least in our view. Last not least, as opposed to the figures, HESS journal may well choose to print the Tables on in different tones of shaded grey. The colours in this table may therefore have to be adapted anyway under the final layout.

[Figure]
 *New, unified colour code for both figures (Fig. 1 & 2)*

Clemens Messerschmid, Amjad Aliewi, Dec 2021

**Additional discussion on the role of slopes:**

In general, many different types of physical characteristics can be used under our approach. As so often, the "art" here is how to keep it as simple as possible[1] while yet simultaneously as differentiated (complicated) as necessary.

The Research Hypothesis in my PhD-Thesis was that: "*recharge is controlled not only by 'outer' meteorological variations in rainfall (and evaporation), but also and especially by land-intrinsic physical features, such as bedrock lithology, soils and land forms (which encompass **relief**, soil, natural vegetation, land use and land cover)*" (freidok.uni-freiburg.de/data/174560, p. 36, section 1.5).

In principle, any number of different land forms or other, typical characteristics can and should be accounted for when creating a specific, intrinsic basin classification framework (BCF). The selection and choice, which set of parameters is best used cannot be decided upon in general. Rather, it is intrinsic to the particular study area at hand (see PUB recommendations). Based on the unique nature of each study area, a specific set of "best", most appropriate physical characteristics has to be picked from a large – in principle, *infinite* - number of possible factors. The researcher should ask: Which land features best express and represent the typical hydrological processes of recharge in the area? Which patterns are most pronounced? But also practically: Which features are easiest and most practical to determine, group and classify under field conditions? Etc.

As a matter of fact, at early stages of our work (and field work), we pondered over the question, whether to single out relief forms such as slopes (their steepness or dip, slope forms, exposition etc.), as an individual group for our BCF. But it soon became clear that in our study area, a rather confusing mix of different degrees of slope dipping, slope types, etc. was likely to emerge. (The amount of work did not seem to justify the amount of work necessary.) In addition, we checked but could not find a pronounced correlation between exposition of slopes and typical vegetation (e.g. north-growing plants vs. south-exposed vegetation…). Therefore we did not use slopes as an additional separate group. However, in other areas, slopes (or more generally: relief forms) may well be indicated as promising grouping factor. It should also be noted here, that the separation and attribution of different features to distinct groups in the BCF does not aim at completeness. Under empirical approaches such as ours, the aim should not be a most exhaustive number of parameter sets (as in Radulovič, 2012). Rather, and in line with PUB, a few, simple and easy to handle sets of land forms should be targeted and selected. The main question should be, whether they do faithfully represent the major processes of recharge in the respective catchment.

Nonetheless, as shown above, we did account for slopes (and terraces) in our matrix of typical soil depths. So, to a certain extent, their spatial distribution was accounted for and included in our analysis, though not singled-out as a separate group in the BCF.

Having said that, you are right in principle and we do encourage other researchers to pay attention to typical slope patterns, both, in their field observations and when approaching and setting up their site-specific BCFs for spatial recharge analysis.
* * *
[1] „*In sum, many authors of PUB emphasized an increasing realization of the need for simplification in hydrological models and a common framework for hydrological modelling, such as Sivakumar et al. (2015), Grayson and Blöschl (2000); Woods (2002); Sivapalan et al. (2003b); McDonnell and Woods (2004); Sivakumar (2004b, 2008a); Wagener et al. (2007); Young and Ratto (2009) and Olden et al. (2012).*" (PhD-Thesis, p. 29)

---

## Author Response (AR1)

This is a brief summary of our answer to the Reviews. It gives an overview over our replies and changes in the revised manuscript

**REVIEW # 1**

**General comments**

a) Wadi Natuf, prior to our work, was entirely ungauged.

b) We concur.

c) It is detailed why our paper is a strong departure from the state of the art (novel approach), with reference to the papers suggested by the reviewer. Clarifications were made in the abstract, introduction and conclusions.

d) The Introduction was shortened, condensed and the amount of information reduced in the revised manuscript.

e) Clarification on the state of remaining "ungauged" inside the West Bank (W. Natuf).

f) The method for the quantification of the Recharge Coefficients was explained (s.a. Specific Comments).

**Specific comments**

1) (Abstract) Clarifications were given on the novelty, and the formulation was highlighted that previously, only "*relatively*" little attention had been paid to spatial distribution of recharge. It was explained that our direct approach focuses on surface features (epikarst, but not deep karst); its applicability was clarified. The suggested additional literature was examined and used where applicable. Changes to the manuscript were made accordingly. (Additional detailed discussion in a separate file was offered, if requested.)

2) (P. 1/26) A comparable quantification (in mm/a) was added to the manuscript

3) (P. 1/28) Another clarification on the novelty of our approach was added.

4) (P.2/5) Language changes were added.

5) (P.2/6) Clarifications and explanation on the nature and state of the aquifers in the Levant were given to the reviewer (no changes in manuscript).

6) (P. 2/36) Changes to the language were made in the manuscript.

7) (P. 3/7) Justification of our terminology and clarification of our reasons (as in used literature). Some but not complete changes to the manuscript (repetitions unavoidable.)

8) (P. 3/9) Some justification of our repetition, but also changes to the manuscript.

9) (P. 3/6) Reasons, why the suggested addition ("fractured density") is not applicable in PUB.

10) (P. 3/20) Obsolete, because Introduction was shortened and the sentence (on Seibert's rainfall-runoff model) was removed from the manuscript.

11) (P. 3/42) The suggestion was considered and the sentence changed in the (revised and shortened) Introduction.

12) (P. 4/9) Suggestion was taken and manuscript changed accordingly (on the WAB aquifer complex and the aquitards).

13) (P. 4/19) Changes to the sentence on Israeli wells were made in the manuscript (clarifications given to avoid the misunderstanding).

14)    (P. 4) Some of the requested changes were made in the revised manuscript (reference and quantification added), but not in full (concerning an early mention of our results already in Ch. 1).

15)    (P. 4/51) Explanation that and why we cannot refer to Hrachowitz (or other literature) in this sentence on our approach – since it is novel and was not mentioned before by other authors. Therefore only partial change of the manuscript.

16)    (P. 5/6) Year corrected in manuscript.

17)    (P. 5/15) Explanation of our criteria – see also changes in our new version of the Introduction.

18)    (P. 5/34) Clarification, justification and proof (that our chalk is indeed impermeable).

19)    (P. 5/36) Slight adjustment of our manuscript (concerning Fig. 2b).

20)    (P. 6/5) The comment by the reviewer was not fully understood. Answers to the different possibilities of what he meant…

21)    (P. 6/21) Confirmation to the reviewer and reference to more details (on perennial springs) in the first article of our series.

22)    (P. 8/44) Partial change to the manuscript regarding the concern voiced by the reviewer (avoiding repetition).

23)    (P. 9/10) Manuscript changed ("mineralogical composition").

24)    (P. 9/12) Manuscript changed (list of karstic features studied in the field).

25)    (P. 10/Tab. 2) Bottom line of Table 2 removed and text in line 18 amended.

26)    (P. 10/15) Manuscript was changed to avoid the understandable confusion about factors and groups in the BCF, and reference to the first article added.

27)    (P. 10/21) Manuscript changed and explained more simply (w/o unnecessary details).

28)    (P. 10/19) Major change of the manuscript (see also comments by Reviewer # 2). The text was changed and Table 3 strongly modified to make our approach better understandable. (Our approach and results themselves remained unchanged of course). – New Table 3.

29)    (P. 10/24) Reference was added to Table 3.

30)    (Ch. 4 Results) Not entirely clear, what the reviewer wanted. But Fig. 5 was changed as suggested (now showing results of all 3 runs, i.e. for all three groups).

31)    (P. 12/6) Recommended change to the manuscript was made ("aquifer").

32)    (P. 13/9) Reduced amount of quoted literature in Ch. 5 Discussion (matter of style) – see also comment 34) below.

33)    (P. 13/21) Exchanged the term ("moderate"), as requested.

34)    (P. 13/24) Clarified that the above passage (comment 32) referred to the existing literature (as bottom line of our discussion).

35)    (P. 13/33 ff.) The paragraph was shortened drastically as requested (see also above comments No. 32 & 34).

36)    (P. 14/5) The above major changes in the paragraph also concern "equifinality", and as mentioned above (comment 30), Fig. 5 was modified (3 maps).

37)    (P. 14/26) The statement was changed (shortened) as requested.

38)    (Section 5.1) An entire section "5.2 Limitations and caveats" was added.

39)    (Section 5.2)  Some of the suggested "Literature on RC values in other areas…" was used and quoted in the revised manuscript (in both, text and references).

40)    (P. 14/16) As requested, the term ("lithology") was replaced.

Answers and manuscript revisions concerning:

**REVIEW # 2**

**General comments**

1. (*Introduction*) Already commented by Review 1. The chapter was reorganised and shortened. Wadi Natuf, prior to our work, was entirely ungauged.
2. (*Quantification of RC-values*) Also, already commented by Review 1 and major changes made in the text and Table 3 of the manuscript.
3. (*Societal issues*) The issue was picked up and the manuscript changed (Section 1.3).
4. (*Climate change & applicability to Mediterranean region*) Both issues were addressed and added to Ch. 6 Conclusions.

**Specific comments**

5. (*Length of Ch. 1*) As in General Comment # 1 above.
6. (*Application to other peri-Mediterranean sites in Ch. 5*) Eight lines of text were added to the manuscript in Ch. 6 Conclusions.
7. (*Quantification of RC-values*) As mentioned above and in our reply to Review 1, major changes were made in the manuscript (methodology better explained; text & Table 3 in Ch. 4 changed).
8. (*Role of slopes our BCF*) A detailed discussion and answer given on this point: Slopes were considered and played a role in our analysis but did not merit an extra Group on its own in our BCF and alongside the three other groups. It was noted that a brief mention of slopes and plains already exists in the caption of Fig. 3. Otherwise, a slight change in the manuscript was made (another reference to the 'matrix' in Table D1 of our first article).

**Technical corrections**

9. (*Colours in Fig. 1 and 2*) The colours were adapted and unified.
10. (*Colours in Table 1*) An answer and justification was given, why the colours in Tab. 1 were not unified with those in Fig. 1 + 2. Also, in the Final layout, the matter of colour in Tables will have to be dealt with, anyway (maybe all in shaded grey?)

Clemens Messerschmid, Amjad Aliewi, Dec 2021